# Distinct myocardial lineages break atrial symmetry during cardiogenesis in zebrafish

Almary Guerra[1†], Raoul FV Germano[1,2†], Oliver Stone[1], Rima Arnaout[3], Stefan Guenther[4], Suchit Ahuja[1], Verónica Uribe[1], Benoit Vanhollebeke[2], Didier YR Stainier[1], Sven Reischauer[1*]

[1]Department of Developmental Genetics, Max Planck Institute for Heart and Lung Research, Bad Nauheim, Germany; [2]Laboratory of Neurovascular Signaling, Department of Molecular Biology, ULB Neuroscience Institute, Université libre de Bruxelles, Bruxelles, Belgium; [3]Division of Cardiology, Department of Medicine, Cardiovascular Research Institute, University of California, San Francisco, San Francisco, United States; [4]ECCPS Bioinformatics and Deep Sequencing Platform, Max Planck Institute for Heart and Lung Research, Bad Nauheim, Germany

**Abstract** The ultimate formation of a four-chambered heart allowing the separation of the pulmonary and systemic circuits was key for the evolutionary success of tetrapods. Complex processes of cell diversification and tissue morphogenesis allow the left and right cardiac compartments to become distinct but remain poorly understood. Here, we describe an unexpected laterality in the single zebrafish atrium analogous to that of the two atria in amniotes, including mammals. This laterality appears to derive from an embryonic antero-posterior asymmetry revealed by the expression of the transcription factor gene *meis2b*. In adult zebrafish hearts, *meis2b* expression is restricted to the left side of the atrium where it controls the expression of *pitx2c*, a regulator of left atrial identity in mammals. Altogether, our studies suggest that the multi-chambered atrium in amniotes arose from a molecular blueprint present before the evolutionary emergence of cardiac septation and provide insights into the establishment of atrial asymmetry.
DOI: https://doi.org/10.7554/eLife.32833.001

**\*For correspondence:**
sven.reischauer@mpi-bn.mpg.de

[†]These authors contributed equally to this work

## Introduction

The mature mammalian heart comprises four chambers that serve the systemic and pulmonary circuits. The parallel arrangement of the two vascular systems was a prerequisite for the evolution of tetrapods from their aquatic ancestors. Intriguingly, the emergence of the four-chambered heart of amniotes from the primitive two-chambered heart of bony fish appears not to have been a single event by axis duplication, but gradually as evidenced by the partial septation of lungfish atria and the complete septation of amphibian atria (*Olson, 2006*).

The formation and patterning of the tetrapod heart relies on the precise integration of cells from distinct progenitor populations during cardiogenesis. These progenitor pools include cells from the bilateral cardiogenic mesoderm, proepicardial progenitor cells, and cardiac neural crest derived cells. Progenitors from the cardiogenic mesoderm are further grouped into the so called first and second heart fields (FHF and SHF, respectively) (*Buckingham et al., 2005*). The integration of these spatiotemporally and molecularly distinct cellular fields is key in cardiac development, as their contributions increase heterogeneity by the addition of distinct cell types, and also define morphogenetic boundaries that shape the heart. Lineage tracing in mammals has revealed that the FHF forms most of the left ventricle, whereas the SHF contributes to the right ventricle and outflow tract

(*Buckingham et al., 2005*; *Simões-Costa et al., 2005*; *Koshiba-Takeuchi et al., 2009*; *Jensen et al., 2013*) thereby establishing a left-right organ asymmetry in the ventricle that is important during ventricular septation and morphogenesis. In contrast to the ventricular chambers, both atria contain cells from both heart fields, but exhibit discrete gene expression programs. Notably, the *Paired-like homeodomain transcription factor 2* c (*Pitx2c*) gene is exclusively expressed in the left atrial myocardium in amphibians and mammals where it regulates a wide variety of genes with left atrial expression (*Figure 1a*) (*Ryan et al., 1998*; *Franco et al., 2014*). Taken together, apparently distinct mechanisms of cellular heterogeneity establish ventricular and atrial laterality in mammals to allow the systemic and pulmonary compartments of the heart to be physically and molecularly distinct. Consequently, mutations interfering with FHF or SHF development or atrial asymmetry are associated with cardiac patterning and septation defects in humans and constitute the leading cause of congenital disorders (*Hoffman, 1995*; *Lin et al., 2012a*; *Li et al., 2015*).

Several transcription factors including Tbx5, Gata4, Nkx2.5 and Isl1 have been implicated in controlling lineage contribution from the FHF and SHF (*Francou et al., 2013*; *Ang et al., 2016*). However, it remains enigmatic how atrial left-right asymmetry is established in the developing tetrapod heart, in part because of the limitations posed by mammalian model systems when it comes to in vivo imaging. Despite its simple two-chambered heart, the zebrafish shows a surprisingly conserved cardiac ontogeny (*Staudt and Stainier, 2012*), including a SHF that contributes to the outer curvature of the ventricle, the outflow tract and atrium (*Zhou et al., 2011*; *Witzel et al., 2012*; *Guner-Ataman et al., 2013*). Hence, studies of cardiac development in zebrafish have contributed significantly to our understanding of the mechanisms underlying lineage diversity and spatiotemporal differentiation (*Witzel et al., 2012*; *Guner-Ataman et al., 2013*; *Stainier et al., 1993*; *Lee et al., 1994*; *Mosimann et al., 2015*; *Rydeen and Waxman, 2016*). Whether the single zebrafish atrium also shows myocardial asymmetry along its left-right axis has not been investigated thus far. Here, we systematically address this question using transcriptomic approaches, combined with the generation of transgenic reporter lines, high-resolution confocal microscopy, lineage tracing, as well as knock-down and loss-of-function experiments. We describe for the first time the existence of two different transcriptional compartments in the atrium of the zebrafish heart, which are analogous to the left and right atria of mammals. Furthermore, we show that this asymmetry appears to be established from two distinct antero-posterior fields during embryonic heart development. We find that the homeobox transcription factor Meis2b is important to establish and maintain left-right asymmetry in the zebrafish atrium, and that it regulates atrial morphogenetic growth and conduction. Altogether, our findings suggest that the multi-chambered heart in mammals evolved from transcriptional compartments present in the piscine heart.

## Results

### Identification of asymmetrically expressed transcription factor genes in the zebrafish atrium

We hypothesized that the four-chambered heart evolved from preexisting cellular compartments rather than the introduction of new structures. To test this hypothesis, we examined in zebrafish atrial-specific transcription factor genes which could potentially be involved in establishing compartments across the atrial left-right axis (*Figure 1b*). Microarray analysis revealed a strong enrichment of *pitx2* transcripts in the zebrafish atrium, reflecting the principal expression domain of its orthologues in the three- and four-chambered hearts of amniotes (*Ryan et al., 1998*; *Campione et al., 1999*; *Tessari et al., 2008*) (*Figure 1b–d*). In humans, *PITX2* mutations can cause Axenfeld-Rieger syndrome and are associated with defects in atrial conduction and septation (*Mammi et al., 1998*; *Gudbjartsson et al., 2007*). In mice, *Pitx2* deficiency causes atrial and ventricular septal defects, right ventricular and right atrial enlargement, and there is strong evidence that *Pitx2* is also key in maintaining left-right atrial identity and sinoatrial node formation (*Franco et al., 2014*; *Tessari et al., 2008*). In addition to *pitx2*, we detected a high atrial enrichment of the *myeloid ecotropic integration site 2b* (*meis2b*) transcription factor gene. This gene was an interesting candidate, as mutations in its human orthologue, *MEIS2*, are strongly implicated in atrial and ventricular septation defects (*Louw et al., 2015*). *Meis2*-deficient mice display outflow tract malformations and persistent truncus arteriosus (*Machon et al., 2015*), while zebrafish

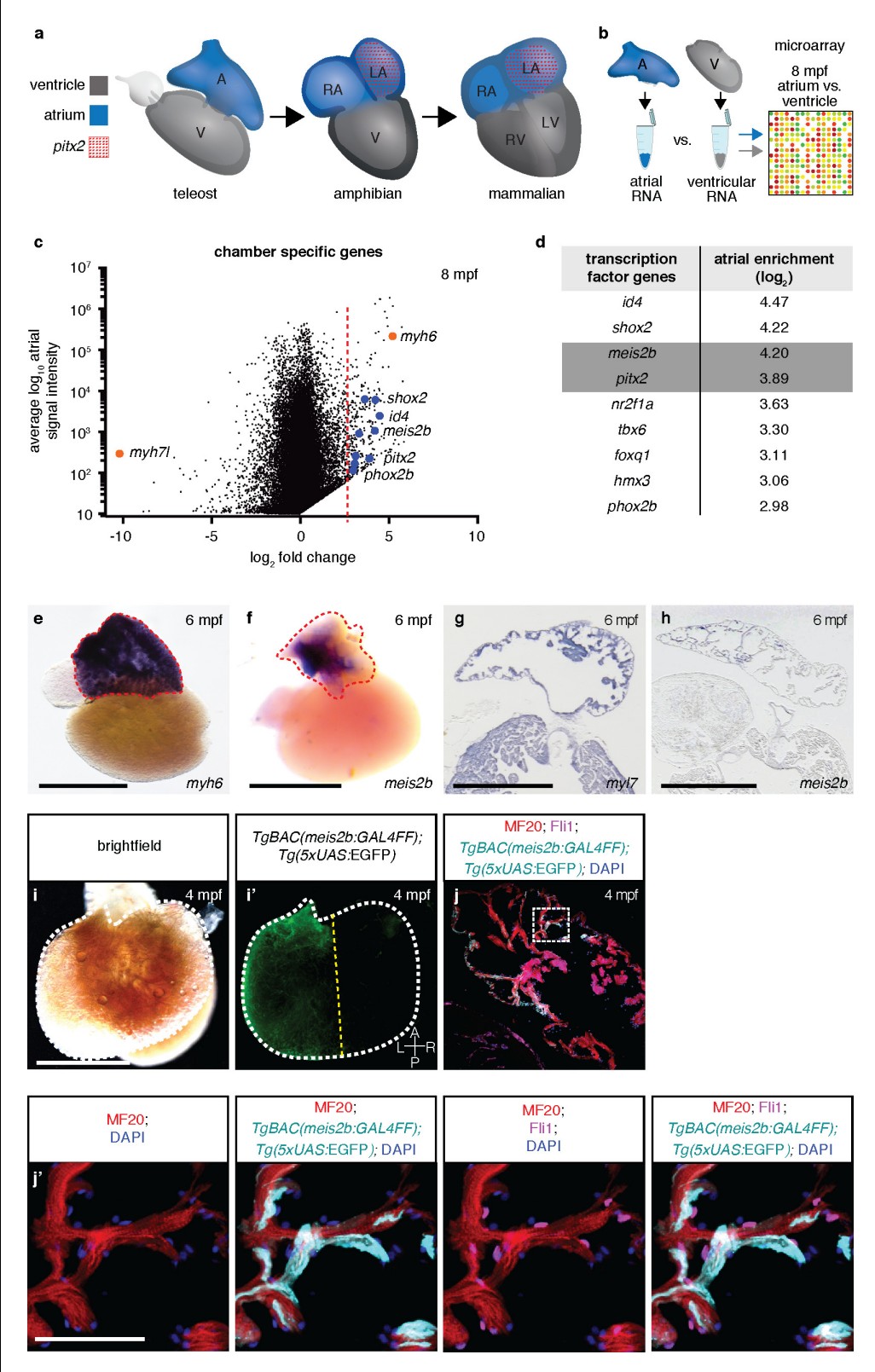

**Figure 1.** Asymmetric expression of atrial-specific transcription factor genes in adult zebrafish. (**a**) Comparison of different vertebrate hearts showing the emergence of distinct left and right chamber identities and atrial septation. (**b**) Schematic representation of the workflow for the identification of cardiac chamber-specific transcription factor gene expression. (**c**) Atrial gene expression level (signal intensity) and chamber-specific

*Figure 1 continued*

enrichment in the atrium (fold change >0) or ventricle (fold change <0). Red line marks 2.5 (log$_2$) fold cutoff. Highly enriched atrial transcription factor genes (blue), ventricular (*myh7l*) and atrial (*myh6*) chamber-specific myosin genes (orange) are shown. (**d**) Atrial enriched transcription factor genes (log$_2$ fold change >2.5). (**e–f**) Whole-mount in situ hybridization on adult zebrafish hearts; *myh6* expression covers the entire atrium while *meis2b* expression is restricted to the left side of the atrium. (**g–h**) In situ hybridization on adult cardiac sections; *myl7* is expressed in all cardiomyocytes while *meis2b* is expressed on the left side of the atrium in the wall and trabeculae (see *Figure 1—figure Supplement 1a* for higher magnification). (**i–i'**) Brightfield and fluorescence images of a 4 mpf *Tg(meis2b-reporter)* zebrafish atrium. (**j–j'**) High-magnification confocal projections of a 4 mpf *Tg(meis2b-reporter)* heart showing expression of the *Tg(meis2b*-reporter) in the myocardium (*Tg(meis2b*-reporter)+/MF20+/Fli cells) but not the endocardium (*Tg(meis2b*-reporter)-/MF20-/Fli1+cells). (**e–f, i–i'**) Red and white dotted lines delineate the atrium; (**i'**) yellow line delineates *Tg(meis2b*-reporter) expression. Scale bars: (**e–i**) 1 mm, (**j'**) 50 μm.

DOI: https://doi.org/10.7554/eLife.32833.002

The following figure supplement is available for figure 1:

**Figure supplement 1.** Validation of the *meis2b* reporter line.

DOI: https://doi.org/10.7554/eLife.32833.003

---

*meis2b* morphants reportedly show cardiac looping defects and a decreased heart rate (*Paige et al., 2012*). Analysis of chromatin remodeling further indicates a role for *MEIS2* during myocardial differentiation of human embryonic stem cells (*Paige et al., 2012*). Interestingly, using in situ hybridization, we found asymmetric expression of *meis2b* in the adult zebrafish heart, as it is exclusively expressed in the left compartment of the mature atrium (*Figure 1e–h*, *Figure 1—figure Supplement 1a*). To investigate *meis2b* expression in detail using live imaging, we established a transgenic reporter system (*TgBAC(meis2b:GAL4FF)$^{bns15}$;Tg(5xUAS:EGFP)$^{nkuasgfp1a}$*) [abbreviated *Tg(meis2b-reporter)*], which recapitulates the embryonic and adult *meis2b* expression as assessed by in situ hybridization (*Figure 1i–i'*, *Figure 1—figure Supplement 1a–f*). Notably, *Tg(meis2b-*reporter) expression is restricted to the left compartment of the adult zebrafish atrium, where it is expressed in the myocardial wall and trabecular myocardium but not expressed in other cardiac cells (*Figure 1j–j'*).

## Left atrial compartments share gene expression programs in fish and mammals

To better understand the myocardial compartmentalization across the atrial left-right axis in the *meis2b* reporter line, we compiled an expression profile of the *Tg(meis2b-*reporter)-positive and -negative domains. We dissected the respective domains in individual hearts (n = 6), isolated RNA and identified the differentially expressed genes (*Figure 2a*). Interestingly, we found a number of genes whose expression showed significant enrichment in the *Tg(meis2b-*reporter)-positive domain (*Figure 2b*). Notably, besides *meis2b* itself, the five most significantly enriched genes were the Pentraxin-related protein gene *ptx3a*, the mitochondrial creatine kinase encoding gene *ckmt2a*, a muscle-specific heat-shock protein *hspb6a*, a zebrafish orthologue of the Norepinephrine transporter encoding gene *Slc6a2*, *si:ch211-117c9.5*, and importantly *pitx2c*, a marker of left atrial identity in mammals (*Figure 2b*). If the *meis2b*-positive left atrial compartment of zebrafish shares an evolutionary origin with the left atrium of mammals, we speculated that genes with left atrial expression in zebrafish would show left atrial enrichment in mammalian hearts as well. We were able to identify direct mammalian orthologues of the five most significantly enriched zebrafish genes and investigated their expression in a previously published expression dataset for atrial asymmetry in mice (*Kahr et al., 2011*). Indeed, we found that all five orthologues were robustly expressed in the mammalian heart and showed strong enrichment within the group of genes with significantly higher expression in the left atrium of mouse hearts (*Figure 2c*). To validate these results, we isolated RNA from mouse left and right atria and, by performing RT-qPCR analysis, found that the expression of *Pitx2c*, *Slc6a2*, *Ckmt2* and *Hspb6* was significantly enriched in the left atrium (*Figure 2d–h*). Notably, we further found that in *Xenopus tropicalis*, the expression of *Pitx2c* was also significantly enriched in the left atrium compared to the right one (data not shown). As *Pitx2c* serves as a marker of left atrial identity in mice, we asked whether *pitx2c* and *meis2b* share a common expression domain within the zebrafish atrium. Hence, we performed

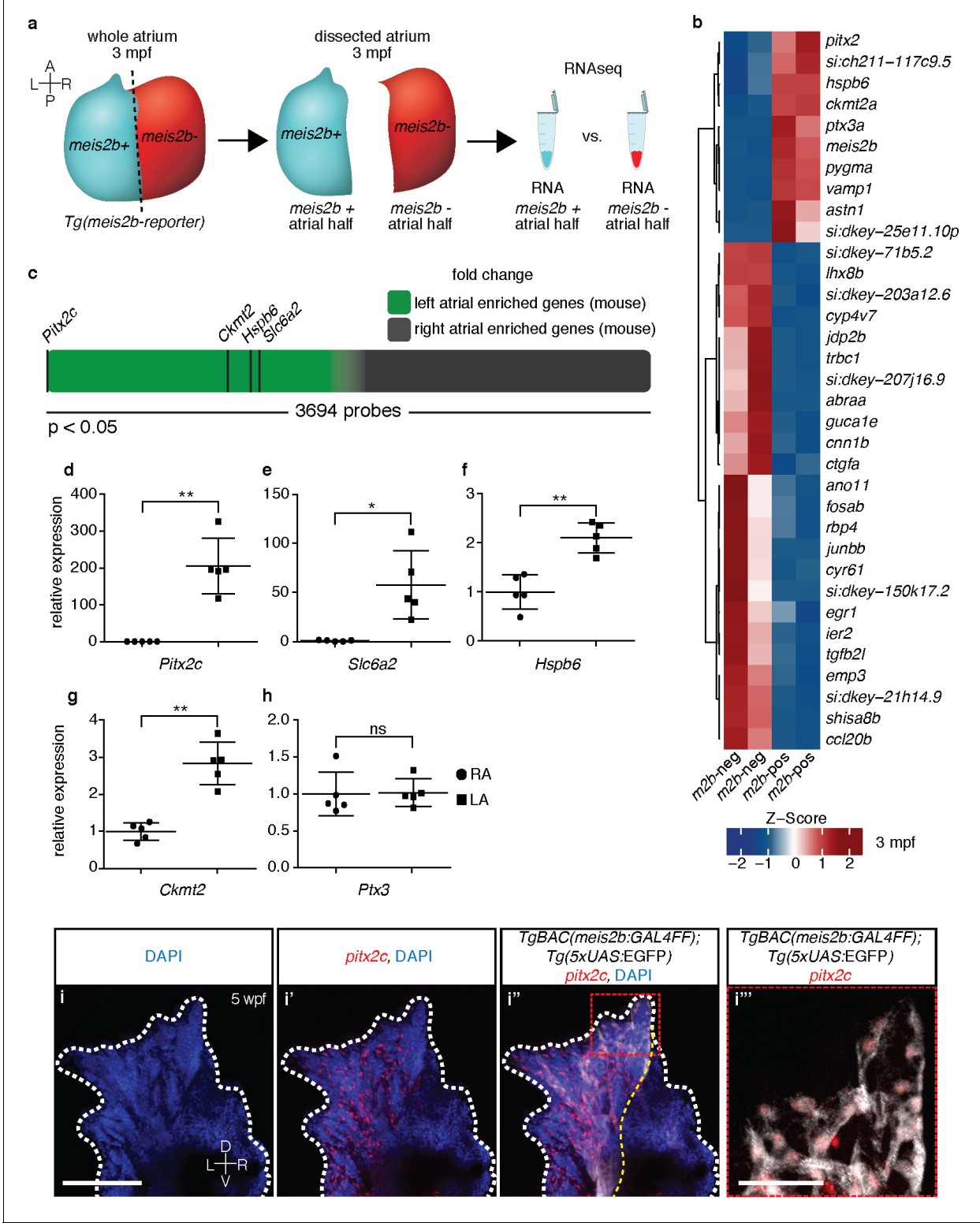

**Figure 2.** The zebrafish atrium is divided into two transcriptionally distinct domains. (a) Schematic illustrating the dissection of a *Tg(meis2b-reporter)* atrium into *Tg(meis2b*-reporter)-positive and -negative domains for RNAseq analysis. (b) Global heatmap depicting Z-score of differentially expressed genes between *Tg(meis2b*-reporter)-positive and -negative domains (*m2b*-pos and *m2b*-neg, respectively). Red color indicates higher expression and blue color lower expression in the respective sample listed at the bottom of each column. (c) Schematic representation of asymmetrically expressed transcripts in mouse atria as determined by microarray analysis (*Kahr et al., 2011*); only significantly differentially expressed genes included (3695/

*Figure 2 continued on next page*

*Figure 2 continued*

25697 probes). Green: significantly left atrial enriched transcripts; grey: significantly right atrial enriched transcripts. (**d–h**) RT-qPCR analysis of left vs. right atria in adult mouse shows that the orthologues of the most significantly enriched genes in the *Tg(meis2b*-reporter)-positive domain are also significantly enriched in the left atrium in mouse. Plots show relative gene expression in left atrium with respect to right atrium (n = 5). (**i–i'''**) Wholemount fluorescent in situ hybridization (RNAscope) for *pitx2c* expression on a *Tg(meis2b-reporter)* zebrafish atrium [DAPI in blue, *Tg(meis2b*-reporter) in white, *pitx2c* in red]. (**b–c, e**) *si:ch211-117c9.5* is a zebrafish orthologue of *Slc6a2*. (**i–i''**) Red and white dotted lines delineate the atrium; (**i'**) yellow line delineates *Tg(meis2b*-reporter) expression. (**d–h**) Two-tailed student t-tests were performed, *p<0.05; **p<0.005; ***p<0.0005; error bars indicate ±SD. Scale bars: (**i**) 100 μm, (**i'''**) 25 μm.
DOI: https://doi.org/10.7554/eLife.32833.004
The following source data is available for figure 2:

**Source data 1.** Ct values obtained in RT-qPCR experiments for adult mouse heart.
DOI: https://doi.org/10.7554/eLife.32833.005

fluorescent in situ hybridization experiments for *pitx2c* expression in the *meis2b* reporter line and found that *meis2b* and *pitx2c* expression largely overlap in the adult zebrafish atrium and are restricted to the left atrial compartment (*Figure 2i–i'''*).

## Atrial laterality appears to derive from two distinct antero-posterior progenitor fields

Little is known about how atrial laterality is established during development. To determine the developmental origin of the observed asymmetric *meis2b* expression, we first performed in situ hybridization studies. We observed that *meis2b* expression appears bilaterally at the 12 somite stage (ss) within the cardiogenic anterior lateral plate mesoderm (ALPM) but is restricted to its posterior aspect (*Figure 3—figure supplement 1*) (*Paige et al., 2012*). The cells of the bilateral ALPM collectively migrate towards the midline where they fuse to form the cardiac disc (*Stainier et al., 1993*; *Trinh and Stainier, 2004*). Consistently, *Tg(meis2b*-reporter) expression was observed only within the posterior aspect of the cardiogenic ALPM (*Figure 3a*, *Figure 3—figure supplement 1j*). Migration to the midline and fusion of the bilateral fields lead to the formation of the cardiac disc which is strikingly asymmetric based on gene expression. At the 20 to 23 ss, once the cardiac disc is formed, *Tg(meis2b*-reporter)-positive cells form a posterior disc compartment (PDC) establishing a sharp boundary with the anterior part of the disc (ADC) which remains *Tg(meis2b*-reporter) negative (*Figure 3b–c*). At this stage, the endocardial progenitors are located ventral to the myocardium, and pass through the central ring of the cardiac disc to connect dorsally to the aortic arches (*Stainier et al., 1993*; *Bussmann et al., 2007*) (*Figure 3—figure supplement 2*; *Video 1*). Notably, the ADC and PDC are rearranged into a dorso-ventral configuration as the heart tube starts to extend (*Smith et al., 2008*; *Rohr et al., 2008*), with the *Tg(meis2b*-reporter)-positive cells located on the ventral side of the tube (*Figure 3d–e*) (*Figure 3—videos 1–5*). Furthermore, we observed that the *Tg(meis2b*-reporter)-positive PDC comprised 54% to 62% of all cardiomyocytes from 17 to 30 ss (*Figure 3a*, *Figure 3—figure Supplement 2d*).

At 50 and 72 hr post fertilization (hpf), after cardiac looping and chamber ballooning have occurred, the *Tg(meis2b*-reporter)-positive cells reside in the dorsal part of the atrium, parts of the sinus venosus, and a small part of the ventricle near the atrio-ventricular canal (AVC) (*Figure 3f*; *Figure 4a,f–f'*; *Figure 3—video 6*, *Figure 4—video 1*). While *Tg(meis2b*-reporter) expression is switched off in the ventricle between 4 and 6 days post-fertilization (dpf), *Tg(meis2b*-reporter)-positive cells populate the left atrial compartment at this stage (*Figure 4—video 2*). As atrial *meis2b* expression appears to be stable during development, it reveals a persistent left-right asymmetry in the adult atrium

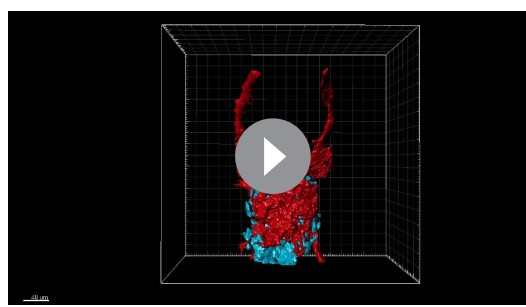

**Video 1.** 3D surface rendering of *Tg(meis2b*-reporter); *Tg(kdrl*:Hsa.HRAS-mCherry) expression at 22 somite stage from *Figure 3—figure supplement 2a*.
DOI: https://doi.org/10.7554/eLife.32833.015

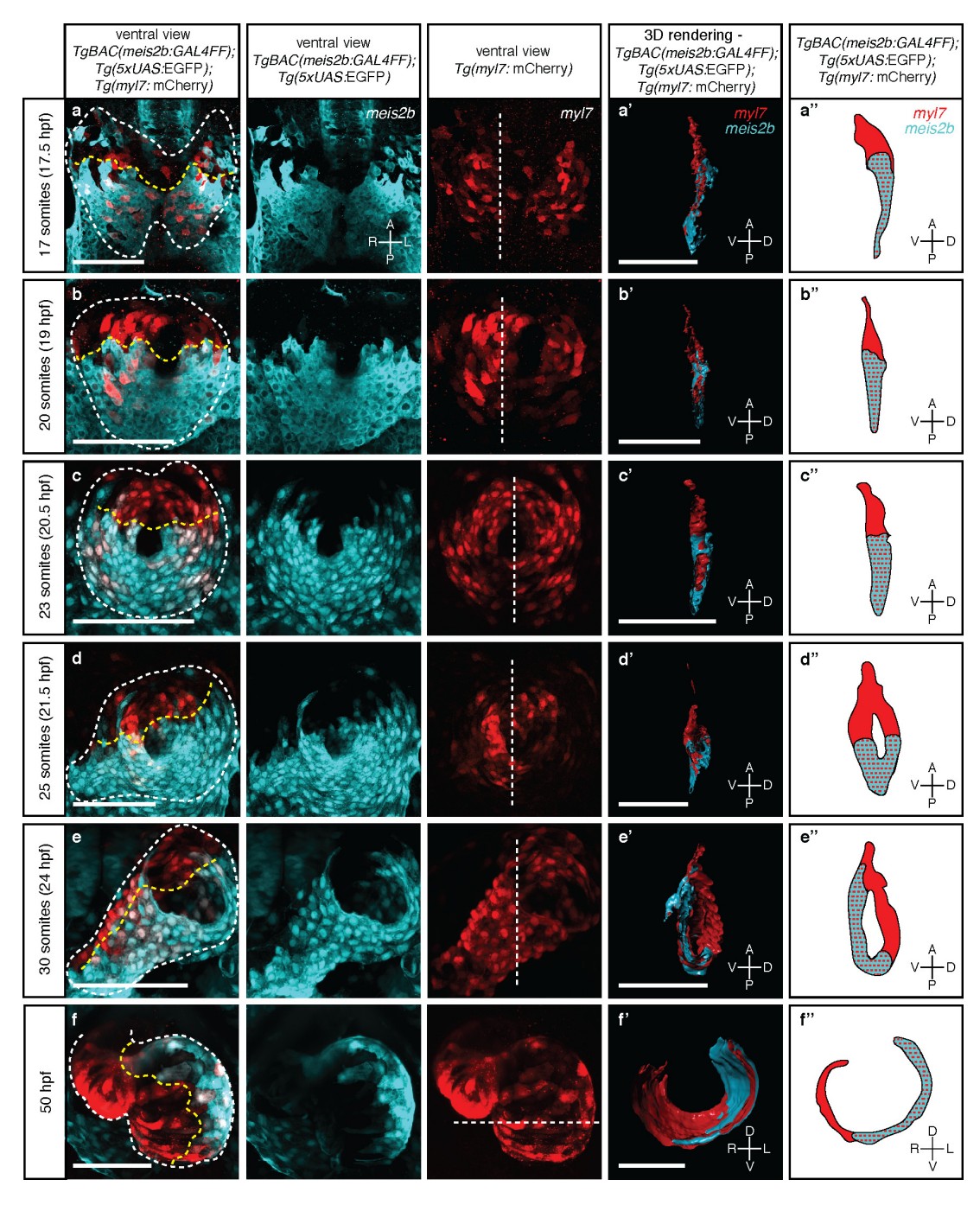

**Figure 3.** Myocardial morphogenesis establishes left-right laterality from two distinct antero-posterior fields. Confocal projections of *Tg(meis2b-reporter);Tg(myl7:mCherry)* embryos between the 17 ss (16 hpf) and 50 hpf. (**a-c**) Myocardial progenitors migrate to the midline and form the cardiac disc, where *Tg(meis2b*-reporter) expression is confined to the posterior compartment of the cardiac disc (PDC). (**d**) Cardiomyocytes forming the cardiac disc appear to migrate clock-wise, rotating the forming heart tube. (**e**) *Tg(meis2b*-reporter)-expressing cells are located on the ventral side of the heart tube. (**f**) At 50 hpf, *Tg(meis2b*-reporter) expression appears on the left side of the atrium, and in a small part of the ventricle near the AV canal. (**a'-f'**) 3D rendered sections (as indicated in the third column, white dotted lines) of the hearts shown in a-f (see *Figure 3—videos 1–6* for full reconstructions). (**a''-f''**) Schematic representation of *Tg(meis2b*-reporter);*Tg(myl7:*mCherry) expression from (**a'-f'**). White dotted lines delineate the heart, yellow dotted lines delineate *Tg(meis2b*-reporter) expression. White dotted lines in third column indicate the level of the sagittal and transverse views. First to third columns: ventral views, anterior up; (**a'-e', a''-e''**) sagittal views, anterior up; (**f', f''**) transverse views, dorsal up. Scale bars: (**a-e'**) 100 μm, (**f'**) 50 μm.

DOI: https://doi.org/10.7554/eLife.32833.006

The following video and figure supplements are available for figure 3:

*Figure 3 continued on next page*

*Figure 3 continued*

**Figure supplement 1.** *meis2b* is expressed in the cardiogenic anterior lateral plate mesoderm.
DOI: https://doi.org/10.7554/eLife.32833.007
**Figure supplement 2.** *Tg(meis2b*-reporter) expression with respect to the endothelium during the cardiac disc and heart tube stages.
DOI: https://doi.org/10.7554/eLife.32833.008
**Figure 3—video 1.** 3D surface rendering of *Tg(meis2b*-reporter);*Tg(myl7*:mCherry) expression at 17 somite stage from *Figure 3a*.
DOI: https://doi.org/10.7554/eLife.32833.009
**Figure 3—video 2.** 3D surface rendering of *Tg(meis2b*-reporter);*Tg(myl7*:mCherry) expression at 20 somite stage from *Figure 3b*.
DOI: https://doi.org/10.7554/eLife.32833.010
**Figure 3—video 3.** 3D surface rendering of *Tg(meis2b*-reporter);*Tg(myl7*:mCherry) expression at 23 somite stage from *Figure 3c*.
DOI: https://doi.org/10.7554/eLife.32833.011
**Figure 3—video 4.** 3D surface rendering of *Tg(meis2b*-reporter);*Tg(myl7*:mCherry) expression at 25 somite stage from *Figure 3d*.
DOI: https://doi.org/10.7554/eLife.32833.012
**Figure 3—video 5.** 3D surface rendering of *Tg(meis2b*-reporter);*Tg(myl7*:mCherry) expression at 30 somite stage from *Figure 3e*.
DOI: https://doi.org/10.7554/eLife.32833.013
**Figure 3—video 6.** 3D surface rendering of *Tg(meis2b*-reporter);*Tg(myl7*:mCherry) expression at 50 hpf from *Figure 3f*.
DOI: https://doi.org/10.7554/eLife.32833.014

(*Figure 4b–c,g–h'*). Together, these findings strongly support the existence in zebrafish of atrial compartmentalization from early stages and throughout life (summarized in *Figure 4d–h'*). To our knowledge, a molecular asymmetry in the single atrium of teleosts has not been described. Altogether, our data provide evidence that asymmetric patterns of gene expression existed in the atrium of fish hearts prior to the evolutionary emergence of the inter atrial septum.

## Atrial asymmetry is established independently of SHF contribution

The establishment of cardiac laterality from two distinct antero-posterior fields has not been described before. To confirm our observations, we lineage traced cells within the cardiac disc by fluorescent labeling and confocal imaging. We injected one-cell stage zebrafish embryos with mRNA encoding KikGR, a photoconvertible green-to-red fluorescent protein, leading to green KikGR expression throughout the developing embryo. After locating the cardiac disc at the 23 ss, photoconversion of the PDC was performed using a confocal laser scanning microscope. Subsequent tracing of the photoconverted red cells revealed their exclusive contribution to the left side of the atrium at 48 hpf, similar to what we observed in the *Tg(meis2b-reporter)* line (*Figure 5a–b''*). Conversely, cells of the ADC contributed to the right side of the atrium at 48 hpf (*Figure 5c–d''*; *Figure 5—videos 1* and *2*). Together, these results reveal a process that establishes atrial left-right asymmetry from an antero-posterior embryonic asymmetry.

The majority of cells that form the early heart tube are derived from FHF progenitors (*Liu and Stainier, 2012*). Subsequently, the SHF contributes cells at its cranial and caudal boundaries (*Guner-Ataman et al., 2013*; *de Pater et al., 2009*; *Hami et al., 2011*). As *Tg(meis2b-reporter)* is expressed in a subset of the atrial myocardium at 48 hpf, we tested whether the *Tg(meis2b-reporter)*-negative atrial myocardium was derived from the SHF. Using immunostainings for the established venous SHF marker Isl1, we tested for a potential inverse correlation between the *Tg(meis2b*-reporter) expression and Isl1-positive cells (*Figure 5e*). As previously reported (*Witzel et al., 2012*), we detected Isl1 expression at the venous pole of the atrium where it is essential for SHF-derived cardiomyocyte differentiation and contribution to the heart tube. However, Isl1 expression did not correlate with *Tg(meis2b*-reporter) expression in the heart. We could detect Isl1 in *Tg(meis2b*-reporter)-positive and -negative cells (*Figure 5e*), suggesting that asymmetric *meis2b* expression is not exclusively associated with the FHF or SHF. However, if a substantial contribution of the SHF was involved in the establishment of a *meis2b*-negative population in the atrium, a knockdown of *isl1* should reduce it significantly. To test this hypothesis, we injected a previously published *isl1* morpholino into the *Tg(meis2b-reporter)* line (*Witzel et al., 2012*). Interestingly, while the injection of *isl1* morpholino resulted in a significant decrease in Isl1 immunostaining, and did not show any obvious toxic effects, the asymmetric expression of *meis2b* within the heart was unaffected at 48 hpf (*Figure 5f*, *Figure 5—figure Supplement 1*; *Figure 5—videos 3* and *4*), indicating that SHF contribution is not required to establish atrial asymmetry. These findings are consistent with our current understanding

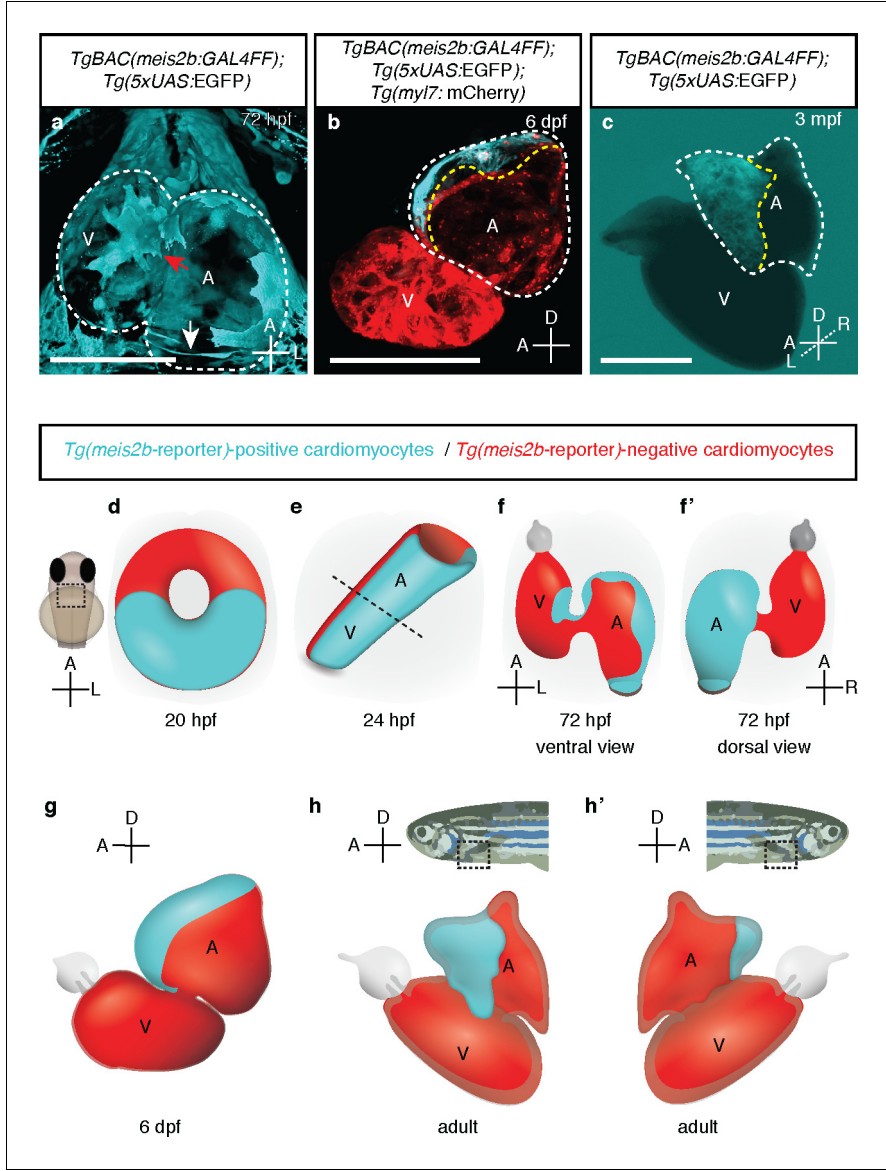

**Figure 4.** *meis2b* expression unveils an atrial asymmetry throughout development. (**a**) At 72 hpf, *Tg(meis2b-reporter)* expression is observed near the sinus venosus (white arrow), the dorsal and distal sides of the atrium, and a small part of the ventricle near the atrioventricular canal (red arrow) (n = 8) (*Figure 4—video 1*). (**b**) At 6 dpf *Tg(meis2b-reporter)* expression is found exclusively in the anterior-left compartment of the atrium at (n = 5) and in adult stages (**c**) in the left side of the atrium (n = 15) (*Figure 4—video 2*). (**d-h'**) Schematic of *Tg(meis2b-reporter)* expression throughout heart development [*Tg(meis2b-reporter)* expression in cardiomyocytes (cyan); *meis2b-*negative myocardium (red)]; (**e**) dotted line indicates the atrioventricular boundary]. (**a, d–f**) ventral views, anterior up; (**b–c, g–h**) lateral views, anterior to the left; (**f'**) dorsal view, anterior up; (**h'**) lateral view, anterior to the right. A, atrium; V, ventricle. (**b–c**) white dotted lines delineate the heart, yellow dotted lines delineate *Tg(meis2b-reporter)* expression. Scale bars: (**a–b**) 100 μm, (**c**) 500 μm.

DOI: https://doi.org/10.7554/eLife.32833.016

The following videos are available for figure 4:

**Figure 4—video 1.** 3D rendering of a 72 hpf *Tg(meis2b-reporter)* heart from *Figure 4a*.
DOI: https://doi.org/10.7554/eLife.32833.017

**Figure 4—video 2.** 3D surface rendering of *Tg(meis2b-*reporter);*Tg(myl7:*mCherry) expression at 6 dpf from *Figure 4b*.
DOI: https://doi.org/10.7554/eLife.32833.018

on the establishment of atrial asymmetry in mammals: in the mammalian heart, while the SHF is important for right ventricular formation, it contributes equally to both atrial chambers and thus is not essential to establish right atrial identity.

## Loss of *meis2b* causes dysmorphic atrial growth and conduction defects

In mammals, atrial asymmetry is important for cardiac septation, morphogenesis and conduction (*Koshiba-Takeuchi et al., 2009*; *Tessari et al., 2008*; *Wang et al., 2010*; *Campione and Franco, 2016*). To determine the role of Meis2b in compartmentalization of the teleost atrium, we induced a frameshift mutation in the highly conserved Hth/Meis domain (*Bürglin, 1997*; *Longobardi et al., 2014*) of *meis2b*, predicted to lead to a truncated protein lacking most functional domains, including

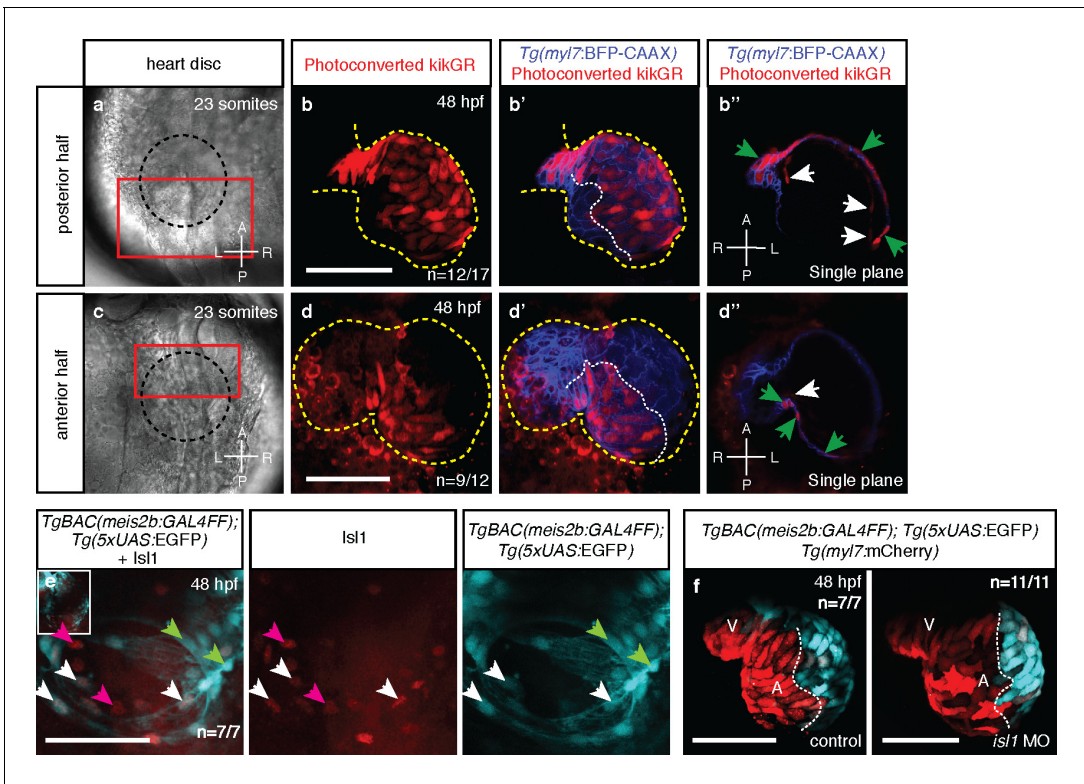

**Figure 5.** Atrial laterality is established from distinct antero-posterior fields without contribution from the second heart field. (**a, c**) Cardiac disc (dotted circle) at 23 ss. (**b–b'**) Confocal projections of photoconverted kikGR with Tg(*myl7*:BFP-CAAX) hearts (yellow lines) show that cells from the PDC populate the left side of the atrium while (**d–d'**) cells from the ADC populate the right side of the atrium (white dotted lines). (**b'', d''**) Single optical planes from b' and d', respectively (cardiomyocytes, green arrows; endocardial cells, white arrows). (**e**) Tg(*meis2b*-reporter) expression and antibody staining for Isl1 in the 48 hpf heart [*meis2b*+/Isl1+ (white arrows), *meis2b*+/Isl1- (green arrows) and *meis2b*-/Isl1+ (pink arrows) cardiomyocytes]. (**f**) Confocal projections of control uninjected and *isl1* MO injected 48 hpf Tg(*meis2b*-reporter);Tg(*myl7*:mCherry) hearts [white lines delineate Tg(*meis2b*-reporter) expression]. Scale bars: 100 μm.

DOI: https://doi.org/10.7554/eLife.32833.019

The following video and figure supplement are available for figure 5:

**Figure supplement 1.** *isl1* morphants have reduced levels of Isl1 protein and phenocopy *isl1* mutants.
DOI: https://doi.org/10.7554/eLife.32833.020
**Figure 5—video 1.** 3D surface rendering of photoconverted kikGR with Tg(*myl7*:BFP-CAAX) expression at 48 hpf from *Figure 5b'*.
DOI: https://doi.org/10.7554/eLife.32833.021
**Figure 5—video 2.** 3D surface rendering of photoconverted kikGR with Tg(*myl7*:BFP-CAAX) expression at 48 hpf from *Figure 5d'*.
DOI: https://doi.org/10.7554/eLife.32833.022
**Figure 5—video 3.** 3D rendering of a 48 hpf Tg(*meis2b*-reporter);Tg(*myl7*:mCherry) heart from *Figure 5f*, control.
DOI: https://doi.org/10.7554/eLife.32833.023
**Figure 5—video 4.** 3D rendering of a 48 hpf Tg(*meis2b*-reporter);Tg(*myl7*:mCherry) heart from *Figure 5f*, *isl1* morphant.
DOI: https://doi.org/10.7554/eLife.32833.024

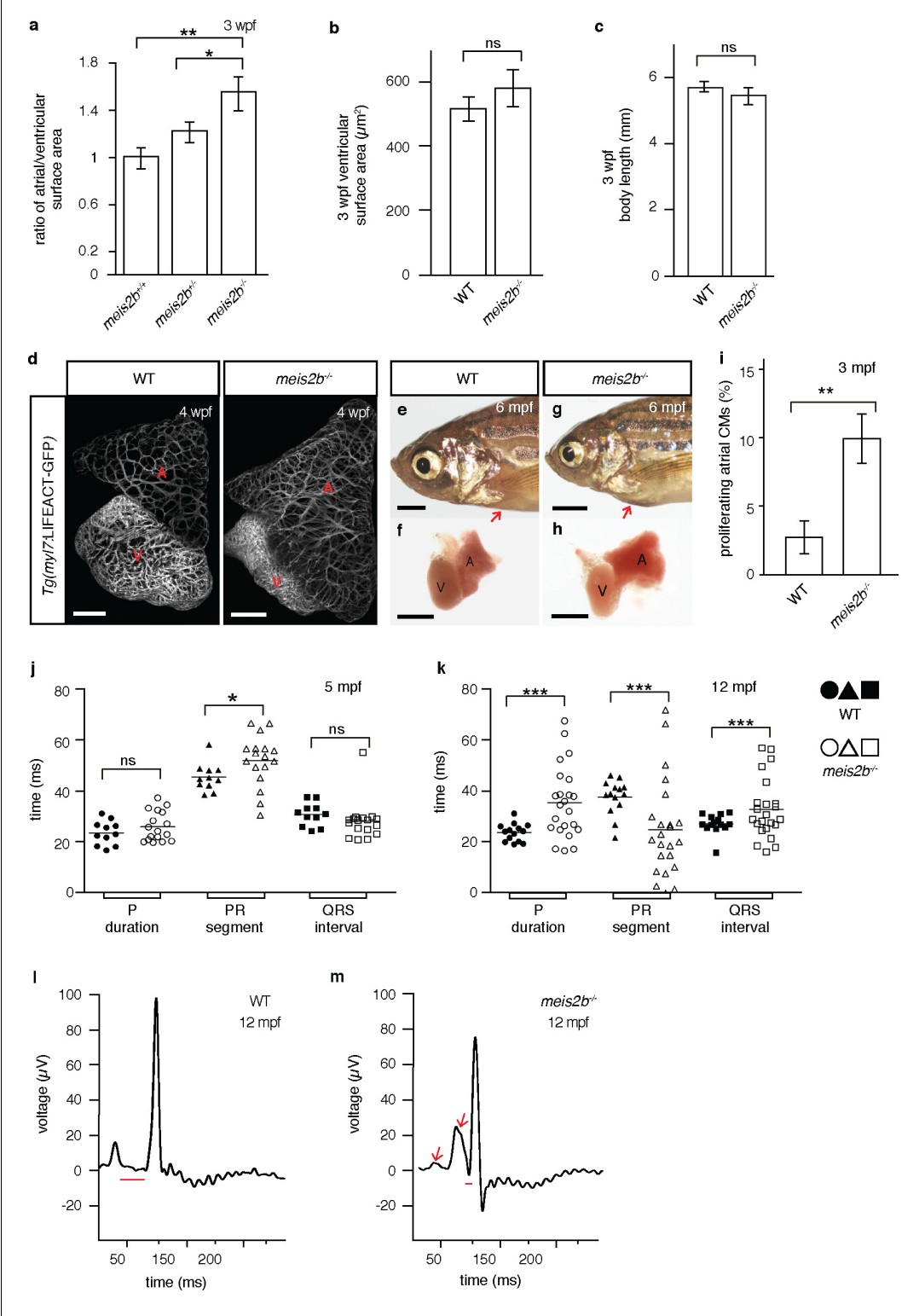

**Figure 6.** Loss of *meis2b* causes dysmorphic atrial growth and conduction defects. (a) *meis2b⁻/⁻* larvae exhibit a significant increase in the ratio of atrial to ventricular surface area, while (b) ventricular surface area and (c) body length are not affected compared to WT siblings. (d) Analysis of *Tg(myl7: LIFEACT-GFP)* fish reveals no obvious defects in atrial myofibrillar architecture in 4 wpf *meis2b⁻/⁻*. (e–h) Adult *meis2b⁻/⁻* display pericardial bulging (arrow) due to abnormal atrial growth. (i) *meis2b⁻/⁻* exhibit a significant increase in atrial cardiomyocyte (CM) proliferation at 3 mpf. (j–k) Cardiac physiology assessed using electrocardiography (ECG) at 5 and 12 mpf. (j) *meis2b⁻/⁻* exhibit a significant increase in PR-segment variability compared to WT siblings, while P-duration and ventricular conduction (QRS) appear unaffected at 5 mpf. (k) At 12 mpf, PR-segment, P-duration and QRS interval are
*Figure 6 continued on next page*

*Figure 6 continued*

significantly affected in *meis2b*[-/-]. (**l–m**) Representative ECGs of 12 mpf animals show reduced PR-segment duration (red line) and multiple P-waves (arrow) in *meis2b*[-/-]. Scale bars: (**d**) 100 μm, (**e–h**) 1 mm. (**a–c, i**) One-tailed student t-tests were performed. (**j–k**) F-test and Bartlett's test were performed. Significant differences compared with control are indicated (*p<0.05; **p<0.005; ***p<0.0005); error bars indicate ± SD. A, atrium; V, ventricle; ns, not significant.

DOI: https://doi.org/10.7554/eLife.32833.025

The following source data and figure supplements are available for figure 6:

**Source data 1.** Quantification of proliferating atrial cardiomyocytes in *meis2b* mutants and WT siblings.
DOI: https://doi.org/10.7554/eLife.32833.028
**Figure supplement 1.** Generation of a *meis2b* mutant allele by TALEN mutagenesis.
DOI: https://doi.org/10.7554/eLife.32833.026
**Figure supplement 2.** Cardiac positioning (jogging) appears unaffected in *meis2b* mutants.
DOI: https://doi.org/10.7554/eLife.32833.027

the DNA-binding homeodomain (*Figure 6—figure supplement 1*). In contrast to previously published knockdown experiments (*Paige et al., 2012*), no early cardiac morphogenetic defect was observed in *meis2b*[-/-], and cardiac jogging did not appear to be affected (*Figure 6—figure supplement 2*). However, as early as 3 weeks post-fertilization (wpf), the ratio of atrial to ventricular surface areas was significantly increased in *meis2b*[-/-] animals, while body length and ventricular size were unaffected (*Figure 6a–c*). Using confocal imaging in the *Tg(myl7:LIFEACT-GFP)* background (*Reischauer et al., 2014*), we assessed atrial myocardial specification and overall myofibril organization. We did not detect architectural defects other than the previously noted overgrowth of the atrium (*Figure 6d*), which results in a dysmorphic atrium and, in some animals, in pericardial bulging (*Figure 6e–h*). As previous reports have suggested an important role for other Meis proteins in the regulation of the myocardial cell cycle (*Mahmoud et al., 2013*), we used EdU incorporation assays in the pan myocardial *Tg(−5.1myl7:nDsRed2)* background to test whether the observed phenotype in *meis2b*[-/-] was a consequence of atrial hypertrophy or increased myocardial proliferation. Strikingly, *meis2b*[-/-] animals showed 10.02% (±1.12) of all atrial cardiomyocytes positive for EdU incorporation after a 4-day exposure in contrast to 2.78% (±0.49) in wild-type (WT) animals, suggesting a major contribution of proliferation to the abnormal atrial growth in *meis2b*[-/-] animals (*Figure 6i*).

We next determined how loss of Meis2b affects cardiac physiology by performing electrocardiographic (ECG) analysis and observed that adult *meis2b*[-/-] frequently display cardiac conduction defects (*Figure 6j–k*). In 5 mpf animals, we found a significantly increased variability in P-R segment duration while P wave duration and ventricular conduction (QRS) appeared unaffected, suggesting a functional defect of the atrial and atrio-ventricular conduction systems in the absence of Meis2b (*Figure 6j*). In 12 mpf fish, these defects became more severe and we observed high variability in P wave duration, extremely long or short P-R segments and, in a few cases, even prolonged ventricular conduction (*Figure 6k*). Most frequently, shortened P-R segments and broad, biphasic P waves (*Figure 6l–m*) were observed (*Weinsaft et al., 2014*). Taken together, these data suggest that *meis2b* controls atrial morphogenetic growth and directly or indirectly the patterning and function of the atrial conduction system.

## Meis2b positively regulates cardiac *pitx2c* expression

Meis proteins evolved from their invertebrate ancestor Homothorax which acts together with Extradenticle (Pbx) and a variety of Hox proteins to regulate the transcription of target genes (*Merabet and Mann, 2016*). In mammals, several splice variants of *MEIS2* have been described including isoforms that lack a C-terminal transactivation domain (TAD), and thus can act as transcriptional repressors (*Hyman-Walsh et al., 2010*). To test whether the zebrafish paralogs *meis2a* and *meis2b* encode transcriptional activators or repressors, we tested their transactivation potential in cultured cells. Using PCR amplification on adult cardiac cDNA, we were able to isolate two different splice isoforms for *meis2a* and one for *meis2b*. Fusions of the C-terminus of all three cardiac *meis2* isoforms to a GAL4-DNA Binding Domain conferred strong transactivation potential (*Figure 7—figure supplement 1*), suggesting that all cardiac Meis2b isoforms function as positive regulators of transcription. To identify potential downstream targets, we next performed expression profiling of whole larvae, entire hearts and dissected atrial chambers of *meis2b*[-/-] and *meis2b*[+/-] siblings at

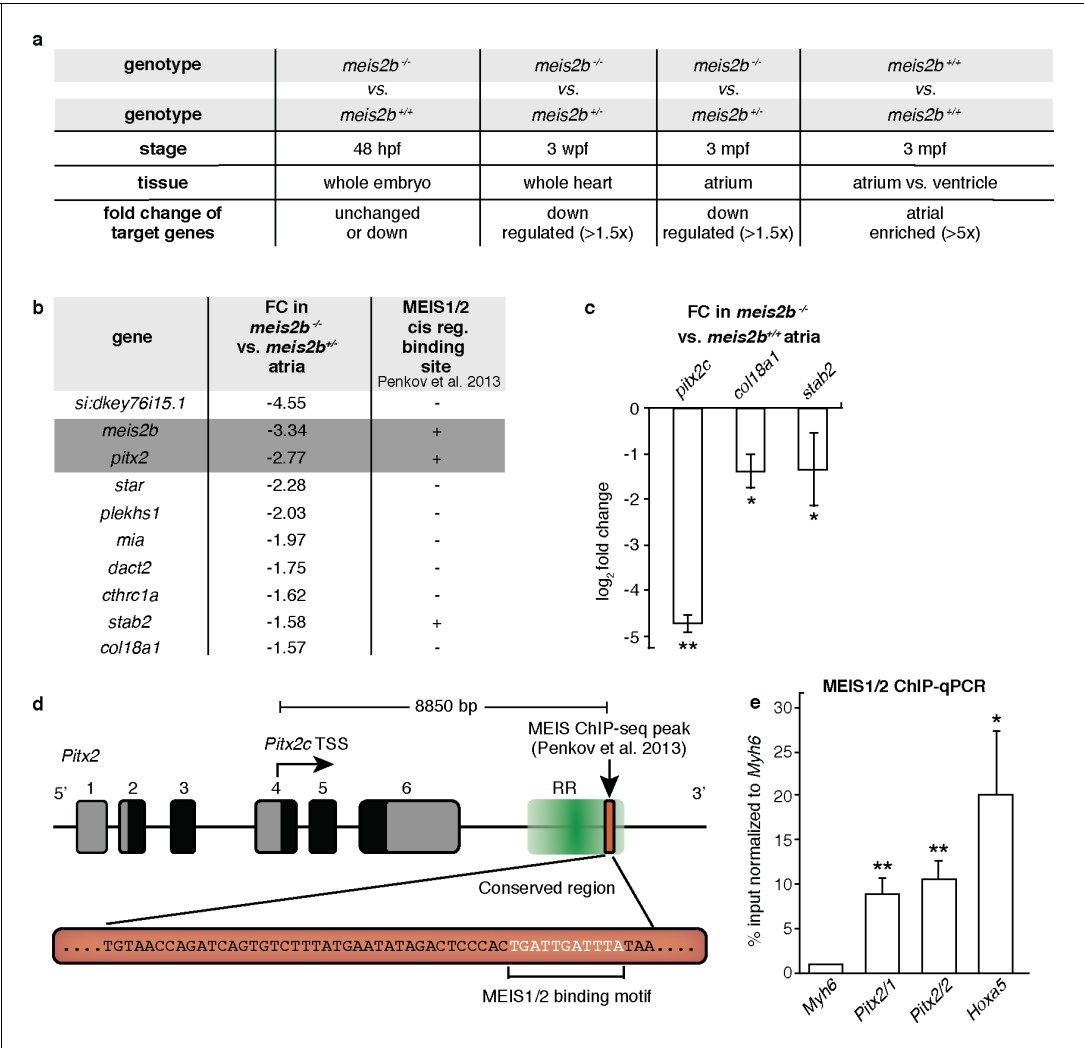

**Figure 7.** Meis2b positively regulates cardiac *pitx2c* expression. (a) Overview of the different samples used for microarray comparison, showing the stage, type of tissue and genotype of the compared samples. The bottom row shows the expected outcome of downstream genes potentially regulated by Meis2b. (b) Table showing the 10 genes fitting all selection criteria (for genes with n > 1, the values are averaged). Cis-regulatory MEIS1/2 DNA binding sites in orthologous mouse genes were extracted from (**Penkov et al., 2013**). (+, proximal binding site detected; -, no proximal binding site detected)]. (c) RT-qPCR analysis in *meis2b*^-/- vs WT sibling atria. (d) Representation of the *Pitx2* locus in mouse indicating the location of a previously reported MEIS1/2 ChIP-seq peak at the 3' end of the regulatory region (RR) (ENSMUSR00000406450) (**Penkov et al., 2013**). A 53 bp conserved sequence inside the ChIP-seq peak is shown together with the MEIS1/2 binding motif. (e) ChIP-qPCR analysis on E12 embryonic mouse trunks of MEIS1/2 on the *Pitx2* locus shown in (d), on the *Hoxa5* locus (a known MEIS1/2 direct target), and on *Myh6* locus (used as a negative control). One-tailed student t-tests (c), and one-sample t-tests (e) were performed, significant differences compared with control are indicated (*p<0.05; **p<0.005); error bars indicate ±SD.

DOI: https://doi.org/10.7554/eLife.32833.029

The following source data and figure supplements are available for figure 7:

**Source data 1.** Ct values obtained in RT-qPCR experiments for adult zebrafish heart.
DOI: https://doi.org/10.7554/eLife.32833.032
**Source data 2.** Ct values obtained in ChIP-qPCR experiments for MEIS on E12 embryonic mouse trunks.
DOI: https://doi.org/10.7554/eLife.32833.033
**Figure supplement 1.** Transactivation potential of Meis2a/b C-terminal domains.
DOI: https://doi.org/10.7554/eLife.32833.030
**Figure supplement 2.** Conserved element inside ChIP-seq peak.
DOI: https://doi.org/10.7554/eLife.32833.031

different developmental stages (*Figure 7a*), resulting in a list of 10 atrial-enriched genes with reduced expression in cardiac tissues of *meis2b*$^{-/-}$ animals at all stages when compared to *meis2b*$^{+/-}$-siblings. Notably, for three of these genes, proximal binding sites for MEIS1/2 have been identified in previous ChIP-seq experiments performed on E11.5 mouse embryos marking them as potential direct targets (*Figure 7b*) (*Penkov et al., 2013*). Notably, the list of genes with reduced expression included *pitx2c*, a key gene of left atrial identity in mammals. Subsequent testing using RT–qPCR confirmed a strong 4.7 (±0.18) log$_2$ fold reduction of *pitx2c* transcripts in adult *meis2b* mutant atria compared to WT siblings (*Figure 7c*).

*Penkov et al. (2013)* performed a ChIP-seq experiment using antibodies that recognize MEIS1 and MEIS2 isoforms. They found in addition to genes in the Hoxa cluster, a region annotated as regulatory (Ensembl, ENSMUSR00000406450), located downstream of the *Pitx2* locus with the closest transcriptional start site belonging to *Pitx2c*. Interestingly, this 187 bp ChIP-seq peak contains a highly conserved 53 bp long element, which includes the MEIS1/2 binding motif (*Figure 7d*, *Figure 7—figure Supplement 2*). To validate these results, we performed a ChIP-qPCR analysis on E12 embryonic mouse trunks using the antibodies employed by *Penkov et al. (2013)*. We performed qPCR on two regions of the *Pitx2* element flanking the MEIS1/2-binding motif, on the *Hoxa5* locus (a known direct target of MEIS1/2 (*Penkov et al., 2013*)), and on *Myh6* (a region where no MEIS1/2 binding site is present). Interestingly, we observed that MEIS1/2 binds *Pitx2* 8.87 (±1.92) to 10.56 (±2.20) fold higher than *Myh6*, while enrichment for the MEIS1/2-binding motif within the *Hoxa5* locus was 20.21 fold (±7.21) (*Figure 7e*). These data suggest that MEIS1/2 can directly bind the regulatory region next to *Pitx2* in mouse. Taken together, our data showing the overlapping expression patterns of *meis2b* and *pitx2* along with the reduced *pitx2c* expression in *meis2b* mutants suggest that *meis2b* acts upstream of *pitx2c* in the heart. These findings are further supported by our ChIP-qPCR data showing that MEIS1/2 directly binds the mouse *Pitx2* locus in vivo as well as the similar *MEIS2* and *PITX2* mutant phenotypes in human and mouse.

## Discussion

Mechanisms that establish cellular heterogeneity during development fulfill two principal roles. First, they increase organ complexity, allowing the emergence of specialized cellular subtypes from a single progenitor field. Second, they define tissue compartments that can serve morphogenetic or functional roles. In the developing vertebrate heart, myocardial cells from the FHF establish a tube that is rapidly compartmentalized into atrial and ventricular domains. Subsequently, a spatiotemporally and molecularly distinct SHF contributes additional myocardial cells to both ends of the tube. At the arterial pole, these cells form the right ventricle, and a septum is formed at the interface between the ventricular descendants of the FHF and SHF (*Buckingham et al., 2005*; *Lin et al., 2012a*). Mutations affecting genes that regulate SHF integration consequently impair right ventricular development and can cause septal defects (*Koshiba-Takeuchi et al., 2009*; *Bruneau, 2008*; *Kelly et al., 2014*). In the venous pole, the second heart field contributes cells to both atria, including functionally important subtypes of cardiomyocytes such as the pacemaker cells and cells that contribute to the atrial septum. The identification of a SHF in zebrafish contributing myocardium to both the ventricle and atrium (*Guner-Ataman et al., 2013*; *de Pater et al., 2009*; *Hami et al., 2011*), was possibly unexpected given the single chamber configuration of its ventricle and the absence of a pulmonary circuit. However, its presence in teleosts suggests that the two-chambered ventricle of amniotes evolved from a preexisting pattern rather than the addition of novel structures. We now report the identification of atrial compartmentalization in the teleost species zebrafish. Moreover, our data suggest that the left atrial compartment in zebrafish shares an evolutionarily conserved origin with the left atrium of tetrapods including the left-specific expression of several genes. Amongst the most significantly left atrial expressed genes in the zebrafish atrium are the transcription factor genes *meis2b* and *pitx2c*, which show overlapping expression patterns in the myocardium of the mature zebrafish heart. *Pitx2c* is a well-established marker of the left atrium in amniotes and a key determinant of left atrial identity (*Tessari et al., 2008*; *Campione et al., 2001*). Furthermore, *PITX2C* mutations have been associated with atrial septation and conduction defects in mammals (*Franco et al., 2014*; *Mammi et al., 1998*; *Gudbjartsson et al., 2007*). *meis2b*, a transcription factor gene of the TALE family is one of two highly conserved zebrafish orthologues of *MEIS2*, a gene that has also been strongly linked to congenital septation defects in humans (*Louw et al., 2015*).

Canonically, embryonic left-right development is regulated through the well studied nodal signaling pathway (*Bakkers et al., 2009*). Using a transgenic reporter line for *meis2b* expression, we were able to track the establishment of atrial asymmetry in zebrafish back to early embryonic stages. Surprisingly, in contrast to components of the nodal pathway, *meis2b* is expressed bilaterally and restricted to the posterior domain of the cardiogenic ALPM. Using in vivo confocal microscopy, we observed that this posteriorly restricted expression of *meis2b* leads to the establishment of two compartments within the cardiac disc, a *meis2b*-positive PDC and an anterior, *meis2b*-negative ADC. Subsequently, a previously reported nodal dependent rotation of the cardiac disc through cell migration breaks lateral symmetry (*Bakkers et al., 2009*), leading to the observed left-right pattern. Using an *isl1* knockdown strategy, we further found that the restricted expression pattern of *meis2b* does not depend upon the integration of cells from the SHF. Meis proteins typically form trimers with their interaction partners, Pre-B-cell leukemia transcription factor Pbx as well as different Hox Proteins, to bind DNA and regulate gene expression (*Merabet and Mann, 2016*). This trimeric mode of action allows highly complex target gene regulation through spatially overlapping transcriptional fields. An important aspect of future studies will be to identify the specific elements that regulate *meis2b* expression and to identify further transcriptional compartments within the cardiac mesoderm to better understand the mechanisms that control cell diversification.

To investigate the role of *meis2b* in cardiac development and asymmetry, we generated a *meis2b* allele predicted to encode a non-functional protein (*Figure 6—figure supplement 1*) (*Longobardi et al., 2014*). Surprisingly, zebrafish *meis2b* homozygous mutants did not exhibit the severe cardiac defects previously reported in a Meis2b knockdown study using splice and translation blocking morpholinos (*Paige et al., 2012*). Notably, maternal zygotic *meis2b* mutants did not exhibit obvious developmental defects either (data not shown). However, *meis2b* homozygous mutants develop atrial overgrowth as well as conduction defects, and these phenotypes become more severe with age. These results suggest that zebrafish *meis2b* does not regulate cardiac looping but atrial growth and conduction.

In mouse, the *meis2b* homologue *Meis1* is important for cardiomyocyte cell cycle regulation, whereby lack of *Meis1* function is sufficient to extend the cardiomyocyte proliferation window in neonatal hearts and induce cardiomyocyte mitosis in adult hearts (*Mahmoud et al., 2013*). Similarly, lack of *meis2b* function in zebrafish could lead to the increase in myocardial proliferation observed in the mutant atria. It will be interesting to further investigate this phenotype including the cell-autonomy of *meis2b* in regulating cardiomyocyte proliferation.

Intriguingly, *pitx2c*, the evolutionary conserved marker for left atrial identity is strongly downregulated in *meis2b* mutants, suggesting that *meis2b* functions upstream of myocardial *pitx2c* expression in zebrafish. Furthermore, we have shown that mouse MEIS1/2 directly binds a conserved regulatory region proximal to the *Pitx2* locus, suggesting a direct regulation of myocardial *Pitx2* expression by MEIS2. These data are particularly interesting, as mutations affecting *PITX2* function have been linked to conduction and septation defects in humans (*Mammi et al., 1998*; *Gudbjartsson et al., 2007*), similar to what has been reported for cases of *MEIS2* haploinsufficiency (*Louw et al., 2015*). Altogether, our findings create new opportunities for the study of atrial compartmentalization and myocardial heterogeneity in zebrafish. It is likely that further investigation of the intrinsic and extrinsic signals that regulate antero-posterior patterning of the cardiac disc will lead to additional insights into the control of cardiac mesoderm diversification and patterning as well as the establishment of atrial asymmetry.

## Materials and methods

### Key resources table

| Reagent type (species) or resource | Designation | Source or reference | Identifiers |
| --- | --- | --- | --- |
| Strain (*Danio rerio*) | *Tg(5xUAS:EGFP)*^nkuasgfp1a | doi: 10.1073/pnas.0704963105 | ZFIN_ZDB-GENO -080528–1 |
| Strain (*Danio rerio*) | *Tg(myl7:mCherry)*^chb1 | doi: 10.1016/j.ydbio.2011.12.004 | ZFIN_ZDB-GENO -110720–1 |

*Continued on next page*

*Continued*

| Reagent type (species) or resource | Designation | Source or reference | Identifiers |
|---|---|---|---|
| Strain (*Danio rerio*) | *meis2b^s988* | This paper | ZFIN: ZDB-ALT-180327-7 |
| Strain (*Danio rerio*) | *TgBAC(meis2b:Gal4FF)^bns15* | This paper | ZFIN: ZDB-ALT-180330-1 |
| Strain (*Danio rerio*) | *Tg(myl7:BFP-CAAX)^bns193* | This paper | ZFIN: ZDB-ALT-180327-8 |
| Antibody | anti-Fli1 Rabbit Polyclonal | Abcam, UK. | ab133485; RRID: AB_2722650 |
| Antibody | anti-MF20 Mouse Monoclonal | Developmental Studies Hybridoma Bank, US. | MF20-s; RRID: AB_2147781 |
| Antibody | Anti-MEIS1/2 | doi: 10.1016/j.celrep.2013.03.029 | Produced at CNIC. K830; K844 |
| Sequence-based reagent | *isl1* ATG morpholino | doi: 10.1242/dev.02355; Gene Tools, LLC, US. | ZFIN_ZDB-MRPHLNO -060728–3 |
| Commercial assay or kit | RNAscope | Advanced Cell Diagnostics, US. | Multiplex Fluorescent kit |
| Software | ZEN Black and Blue 2012 | Zeiss, Germany. | |
| Software | NIS-Elements | Nikon Instruments Inc. | |
| Software | Imaris - Version 8.4.0. | Bitplane, UK. | |
| Software | LabChart 8 | ADInstruments, Australia. | |
| Software | PerlPrimer - Version 7 | doi: 10.1093/bioinformatics/ bth254 | |
| Other | Dynabeads Protein G | Invitrogen, Germany. | 10003D |

## Zebrafish husbandry and strains

All zebrafish husbandry was performed under standard conditions in accordance with institutional (MPG) and national ethical and animal welfare guidelines. The following transgenic and mutant lines were used in this study: *Tg(myl7:LIFEACT-GFP)^s974* (*Reischauer et al., 2014*), *Tg(−5.1myl7: nDsRed2)^f2* (*Mably et al., 2003*), *Tg(5xUAS:EGFP)^nkuasgfp1a* (*Asakawa et al., 2008*), *Tg(myl7:mCherry)^chb1* (*Langenbacher et al., 2012*), *Tg(myl7:EGFP-Hsa.HRAS)^s883* (*D'Amico et al., 2007*), *Tg(kdrl: Hsa.HRAS-mCherry)^s896* (*Chi et al., 2008*), and *isl1^sa0029* (*de Pater et al., 2009*).

## Generation of transgenic and mutant lines

TALEN technology was used to target *meis2b*. TALEN-binding sites were selected using TALEN Targeter 2.0 (tale-nt.cac.cornell.edu) for exon 4 of *meis2b* (NM_130910.1), and constructs were assembled as previously described (*Cermak et al., 2011*; *Bedell et al., 2012*) to bind the target sequences 5'-TTGAGAAGTGCGAGCT-3' and 5'-AGAGAGCCGGGAGTCGCTGGA-3' for the right and left arms, respectively. mRNA encoding the TALEN arms was synthesized using the mMESSAGE mMachine Kit (Ambion), and 100 pg per TALEN arm were injected into one-cell stage embryos to generate potential founders (F0). To assess the efficacy of the TALEN injections, DNA was extracted from injected embryos at 48 hr post fertilization (hpf) and subjected to high-resolution melt analysis (HRMA, Illumina Eco Real-Time PCR System). F0 founders were identified by outcrossing to wild-type (WT) fish. Genomic DNA was extracted from individual F1 embryos and analyzed for induced genomic mutations by PCR amplification of the TALEN target region and subsequent HRMA (*Supplementary file 1a*). Using the genomic DNA of F1 fin clipped adult fish as a template, PCR products were obtained with the primers shown in *Supplementary file 1b* and sequenced in order to determine the specific alleles generated. An allele containing a 10 bp deletion with a 3 bp insertion leading to a frameshift mutation and a premature stop codon was identified and designated *meis2b^s988*. Genotyping of *meis2b^s988* allele was carried out by PCR using allele specific primer combinations (*Supplementary file 1c-d*).

The *TgBAC(meis2b:Gal4FF)^bns15* line was generated by rapid modification of a Bacterial Artificial Chromosome (BAC) clone using a recombination-based protocol (*Bussmann and Schulte-Merker, 2011*). The modified clone (CH211-128L12) contains ~139 kb of genomic sequence upstream of the *meis2b* start codon and ~20 kb downstream its stop codon. Briefly, after Tol2 transposon LTR

insertion into the vector backbone, the recombination was performed by amplifying a Gal4FF-Kan(R) cassette using primers with 50 bp of homology flanking the start codon of *GAL4FF* (*Supplementary file 1e*). Transgenesis was performed in the *Tg(5xUAS:EGFP)* background, following a previously described protocol (*Kawakami et al., 2004*).

Generation of *Tg(myl7:BFP-CAAX)*^bns193^: Blue fluorescent protein (BFP) coding sequence was fused to a CAAX membrane targeting motif (*Lin et al., 2012b*) using Cold Fusion technology (System Biosciences, CA) and cloned under the control of the *myl7* promoter (*Reischauer et al., 2014*) to obtain *myl7:BFP-CAAX*. 20 pg of the resulting plasmid along with 15 pg Tol2 mRNA were injected in one-cell stage embryos to generate *Tg(myl7:BFP-CAAX)*^bns193^.

## RNA in situ hybridization

RNA in situ hybridization experiments on embryos were performed as described (*Thisse and Thisse, 2008*). For whole-mount adult hearts, in situ hybridization was performed with the following modifications: permeabilization of the tissue was achieved by incubating the hearts in proteinase-K (10 μg/mL) for 30 min on ice and then for 40 min at 37°C with gentle agitation.

In situ hybridization on 12-μm-thick paraffin sections of adult hearts: briefly, after rehydration, the slides were incubated with pre-hybridization solution (50% formamide, 5X SSC pH 4,5, 50 μg/mL yeast tRNA, 1% SDS, 50 μg/mL Heparin, 5X Denhardt's solution) for 1 hr at 65°C. Subsequently, the probe was added to a final concentration of 1 μg/mL to hybridize overnight at 65°C. The next day, samples were washed with a series of SSC to MBST (100 mM Maleic acid, 150 mM NaCl, 0.1% Tween 20, pH 7.5) and blocked for 2 hr using MBST with 2% Blocking Reagent (Roche) and 5% sheep serum. Subsequently, anti-DIG-AP Fab fragments (Roche) were added and incubated overnight at 4°C. Staining was performed with BM Purple (Roche). Samples were imaged using a Zeiss Axiocam mrc5 camera or Nikon SMZ25. RNA probes were synthesized from cDNA from 52 hpf zebrafish embryos and adult zebrafish hearts using the combination of primers listed in *Supplementary file 1*.

RNAscope (Advanced Cell Diagnostics, Hayward, CA) was performed on 5 weeks post-fertilization (wpf) *Tg(meis2b-reporter)* hearts, following a published protocol (*Gross-Thebing et al., 2014*) for whole-mount RNAscope. Advanced Cell Diagnostics designed the *pitx2c* probe used in this study.

## Morpholino injections

*isl1* ATG morpholino from Gene Tools, LLC (5'-CCCATGTCAAGAAAGTAAGGCGGTG-3') (*Hutchinson and Eisen, 2006*) was prepared at a stock concentration of 1 mM and diluted to the desired concentration for microinjection. 5 ng of the *isl1* morpholino were injected into one-cell stage *Tg(meis2b-reporter)*;*Tg(myl7:mCherry)*, *Tg(myl7:EGFP-Hsa.HRAS)* and *isl1* mutant embryos. Uninjected embryos and embryos injected with control morpholino served as controls. Embryos injected with control morpholino did not show any detectable phenotype. The embryos were imaged at 48 hpf using a Zeiss LSM 700 confocal microscope.

## kikGR injection and photoconversion

100 pg of *kikGR* mRNA were injected into one-cell stage *Tg(myl7:BFP-CAAX)* embryos. Dechorionated embryos of 20–23 somite stage (ss) were mounted in 0.8% agarose with the dorsal side up. Using a Zeiss LSM 880 confocal microscope, the cardiac disc was identified, and the anterior or posterior half of the disc was selectively photoconverted using a 405 nm wavelength laser (7 pulses of 1 s each and 15% laser power).

After photoconversion, the embryos were transferred to egg water and raised until 48 hpf. Then, the embryos were mounted in 0.8% agarose and the hearts were imaged by confocal microscopy (Zeiss LSM 880). Each heart was imaged twice: first, the green kikGR was imaged together with the photoconverted kikGR. In a second step, *Tg(myl7*:BFP-CAAX) expression was visualized by excitation with a 405 nm wavelength laser. The resulting images containing the different channels were merged using ImageJ.

## Embryonic and juvenile heart imaging and measurements

Offspring of a *Tg(myl7:LIFEACT-GFP);meis2b*$^{+/-}$ incross were grown until 3 and 4 wpf. The juvenile fish were fixed in Fish-Fix (0.1 M phosphate buffer, 4% PFA, 4% sucrose, 120 µM CaCl$_2$) for 1 hr at room temperature. The hearts were dissected and individually mounted in 1% low-melt agarose. After imaging with a Zeiss LSM 780 confocal microscope, the fish were genotyped by PCR using the above-mentioned set of allele specific primers.

Zen Microscope Software (Zeiss, Black Edition 2012) was used to create maximum intensity projections. Ventricular and atrial maximal surface area was defined as the outer boundary of each compartment, as assessed by manually examining each z-stack. ImageJ was used to compute the surface area.

For imaging at early stages, embryos were mounted in 1% low-melt agarose and imaged using a Zeiss LSM 700 confocal microscope. ImageJ was used to count the cells.

For determining the heart rate at 48 hpf, the embryos were embedded in 1% low-melt agarose containing 0.004% tricaine (*Chan et al., 2009*). Videos of 1 min duration were acquired using a Nikon SMZ25 and the heart rate was measured.

Adult zebrafish electrocardiograms were performed as described (*Orr et al., 2016*). Signals were amplified using the Animal BioAmp FE136, acquired using the PowerLab 4/30, and analyzed using LabChart 8, all from ADInstruments (Sydney, Australia). 40 *meis2b*$^{-/-}$ and 25 WT siblings were tested. For each fish, a minimum of 20 heartbeats with a clearly identifiable baseline were used to analyze ECG tracings.

## Image processing

*Figure 5B* was processed with PhotoFiltre seven for noise reduction (pixel dimension, two and threshold, 0). *Figure 4B* was processed using Imaris 8.4.0 for cropping areas outside of the heart and to enhance brightness. Adjustments for brightness and contrasts of other confocal projections and brightfield pictures were performed with Zen Black and Zen Blue editions (Zeiss), NIS-Elements (Nikon) and ImageJ (NIH).

## Gene expression profiling

To identify chamber-specific transcripts, 12 adult zebrafish hearts (8 months post-fertilization, mpf) were isolated as described (*Arnaout et al., 2014*), and bisected into atria and ventricles. The expression profile of the pooled atria was compared to the expression profile of the pooled ventricles. Briefly, total RNA was isolated using TRIzol (Sigma), treated with DNase (Promega, Germany), and purified using an RNA Clean and Concentrator kit (Zymo Research). Gene expression profiles were established using dual color microarray analysis by MOgene (St. Louis, MO) using the Agilent Zebrafish V3 44K platform.

To determine the expression profiles of the *Tg(meis2b*-reporter)-positive vs -negative atrial compartments, a total of 6 hearts of 3 mpf *Tg(meis2b-reporter)* zebrafish were micro-dissected. A total of 4 pools were made: the first two pools each contained 3 *Tg(meis2b*-reporter)-positive atrial compartments, and the other two contained the *Tg(meis2b*-reporter)-negative parts of the atria. RNA was isolated using the miRNeasy micro Kit (Qiagen) combined with on-column DNase digestion (DNase-Free DNase Set, Qiagen) to avoid contamination by genomic DNA. RNA and library preparation integrity were verified with a BioAnalyzer 2100 (Agilent) or LabChip Gx Touch 24 (Perkin Elmer). 50–100 ng of RNA were used as input for Truseq Stranded mRNA Library preparation following the low sample protocol (Illumina). Sequencing was performed on a NextSeq500 instrument (Illumina) using v2 chemistry, resulting in a minimum of 27 M reads per library with a 2 × 75 bp paired-end setup. The resulting raw reads were assessed for quality, adapter content and duplication rates with FastQC. Trimmomatic version 0.33 was employed to trim the reads after a quality drop below a mean of Q18 in a window of 5 nucleotides (*Bolger et al., 2014*). Only reads above 30 nucleotides were cleared for further analyses. Trimmed and filtered reads were aligned versus the Ensembl Zebrafish genome version DanRer10 (GRCz10.87) using STAR 2.4.0a with the parameter '–outFilterMismatchNoverLmax 0.1' to increase the maximum ratio of mismatches to mapped length to 10%. The number of reads aligning to genes was counted with featureCounts 1.4.5-p1 tool from the Subread package. Only reads mapping at least partially inside exons were admitted and aggregated per gene. Reads overlapping multiple genes or aligning to multiple regions were excluded.

Differentially expressed genes were identified using DESeq2 version 1.62. Only genes with a minimum fold change of ±1.5 (log2 ±0.59), a maximum Benjamini-Hochberg corrected p-value of 0.05, and a minimum combined mean of 5 reads were deemed to be significantly differentially expressed. The Ensembl annotation was enriched with UniProt data (release 06.06.2014) based on Ensembl gene identifiers.

For the study of possible downstream targets of Meis2b, we compared the following expression profiles: 48 hpf whole larvae of *meis2b*$^{-/-}$ to *meis2b*$^{+/+}$ siblings, whole hearts of 3 wpf *meis2b*$^{-/-}$ to *meis2b*$^{+/-}$ siblings, whole atria of 3 mpf *meis2b*$^{-/-}$ to *meis2b*$^{+/-}$ siblings, and ventricles to atria of 3 mpf WT zebrafish. Total RNA was isolated using TRIzol and purified using column-based affinity purification (RNA Clean and Concentrator kit, Zymo Research). Gene expression profiles were established using single color microarray analysis OakLabs (Hennigsdorf, Germany) using the ArrayXS Zebrafish 8 × 60K platform (OakLabs).

## RT-qPCR analysis

To study gene expression in the adult mouse heart, the left and right atria of 5 adult hearts were dissected. RNA was isolated using TRIzol and cDNA was synthesized using a Maxima First Strand cDNA Synthesis Kit (Thermo Fisher Scientific) following manufacturer's instructions. Plots show the result of five different experiments (one atrium per experiment). Primers were designed using PerlPrimer (*Marshall, 2004*), and *Actb* was used for normalization. Primer sequences are listed in *Supplementary file 1*. The Ct values obtained for every gene are shown in *Figure 2—source data 1*.

For RT-qPCR expression analysis of potential *meis2b* downstream targets, total RNA was isolated from the atria of 3 mpf (adult) *meis2b*$^{-/-}$ and *meis2b*$^{+/+}$ siblings using TRIzol and cDNA was synthesized as described above. Plots show the result of three different experiments, representing three pools of six atria each. Primers were designed using PerlPrimer (*Marshall, 2004*), and *rpl13* was used for normalization. Primer sequences are listed in *Supplementary file 1*. The Ct values obtained for every gene are shown in *Figure 7—source data 1*.

Of note, reference genes do not show different expression levels between the left and right components of the atria in fish or mice (*Kahr et al., 2011*).

## ChIP-qPCR

ChIP experiments were performed as described (*Penkov et al., 2013*) with minor modifications using WT C57BL/6 E12 embryonic mouse trunks (without head, tail or legs). The protocol was started with approximately 5 × 10$^7$ cells (equivalent to three trunks). The Meis antibodies were kindly provided by the Torres lab (K830, recognizing Meis1a and Meis2a isoforms, and K844, recognizing Meis1a and Meis1b isoforms; both produced at the CNIC, Madrid, Spain).

Briefly, single-cell suspensions were prepared in cold DMEM +10% FBS by crushing embryonic mouse trunks against a 100 µm cell strainer. Cross-linking was performed with 1% formaldehyde (Thermo Fisher Scientific) for 10 min, and the reaction was quenched by addition of 125 mM glycine. The cells were harvested and washed three times with 30 mL cold PBS and finally resuspended in 2.5 mL LB1 for 20 min at 4°C. The nuclei were washed with LB2 for 10 min at 4°C, and resuspended in 3 mL of LB3. At this point, the samples were divided in half to continue the protocol with approximately 2.5 × 10$^7$ cells for each immunoprecipitation. Chromatin was sonicated three times (12 min cycles; 10 s pulse, 50 s rest) at 20% amplitude, using a Bandelin Sonopuls ultrasonic homogenizer to generate 200–1000 bp chromatin fragments.

After centrifugation and verification of the size of the sonicated chromatin, the sonicated lysates were incubated with antibody-bound Dynabeads protein G (Invitrogen). Rabbit IgG IP was used as a negative control. After overnight incubation at 4°C, three buffers were used (WB1, WB2 and WB3) and the immunoprecipitated material was washed twice at 4°C with 1 mL of each buffer. Finally, the elution of the immunoprecipitated material was done at 37°C with TE buffer +2% SDS for 30 min. After removing the magnetic beads, the eluted material was reverse cross-linked overnight at 65°C. Three volumes of TE buffer were added to dilute the SDS in the solution. After treatment with RNase-A and proteinase-K, the DNA was extracted twice with phenol-chloroform and purified with a MinElute PCR Purification Kit (Qiagen). Four biological samples were used for the ChIP. The qPCRs were done in triplicates, using 2X SYBR green real-time PCR master mix (Thermo Fisher Scientific) in

a CFX Connect Real-Time PCR detection system (BioRad) with 39 cycles. Percentage input was calculated using the DDCT method. The list of primers employed for the qPCR can be found in *Supplementary file 1*. The Ct values obtained in the qPCR can be found in *Figure 7—source data 2*.

### EdU injections

Eight *meis2b*[-/-] fish and 8 WT siblings at 3 mpf were anesthetized in 0.02% tricaine. EdU (Life Technologies, # A10044) was injected intraperitoneally (200 µg/g of body weight) and incubated for 4 days. The EdU was detected using a Click-iT EdU Alexa Fluor 647 Imaging Kit (Life Technologies, # C10340).

### Antibody staining and tissue clearing

Whole-mount staining of Isl1 on 48 hpf *Tg(meis2b-reporter)* embryos was performed as previously described (*Dong et al., 2007*). With the exception that the washing solution contained 0.1% Triton X-100 to reduce background.

*Tg(−5.1myl7:nDsRed2)* adult zebrafish hearts were extracted and fixed in 4% paraformaldehyde overnight. Hearts were gradually dehydrated in methanol and the tissue was permeabilized by incubating the hearts for 1 hr at −80°C and then 1 hr at room temperature, repeating this process 6 to 8 times. The hearts were rehydrated in PBS and blocked in PBS + 1% BSA+1% DMSO+0.5% Triton 100x + 2% Goat Serum for 5 hr at room temperature. Rehydrated hearts were incubated for 5 days at 4°C with anti-DsRed Rabbit Polyclonal Antibody (Clontech, # 632496) diluted 1:500 in PBDT (PBS + 1% BSA+1% DMSO+0.5% Triton 100x). After incubation with the primary antibody, the hearts were washed with PBDT and incubated for 5 days at 4°C with Alexa Fluor 488 Donkey-anti-rabbit (Life Technologies, #A21206), diluted to 1:500 in PBDT. At this point, the EdU was detected following the company's protocol.

For antibody staining on cryosections, adult hearts were fixed in 4% paraformaldehyde overnight, and washed with PBS + 0.1% Tween. Fixed samples were incubated with 1 mL of 10% sucrose in PBS (w/v) at room temperature until they sank to the bottom of the tube, and then transferred to 30% sucrose in PBS overnight at 4°C. The samples were mounted in OCT (Tissue-Tek), and 12-µm-thick sections were cut. The sections were washed first in water and then three times in PBS. After blocking the sections (with PBS + 2% sheep serum +0.2% triton X-100 +1% DMSO) for 2 hr at room temperature, the sections were incubated with anti-Fli1 Rabbit polyclonal antibody (ab133485, Abcam. Diluted 1:100), MF20 mouse monoclonal antibody (DSHB, diluted 1:200), Chicken anti-GFP (Aves Labs, GFP-1020. Diluted 1:500) overnight. The samples were washed three times with PBS + 0.1% Triton-X100 and incubated 3 hr with secondary antibodies (goat anti-rabbit Alexa Fluor 647; goat anti-mouse Alexa Fluor 568; goat anti-chicken Alexa Fluor 488, diluted 1:500) at room temperature. Finally, the samples were incubated with DAPI (diluted 1:10,000) in PBS + 0.1% Triton-X100 for 5 min, mounted in Mowiol (Sigma) and imaged with a Zeiss LSM 800 confocal microscope.

The tissues were cleared using iDISCO (*Renier et al., 2014*) and imaged using a Zeiss LSM 800 confocal microscope. IMARIS was used to count the number of cardiomyocyte nuclei and the number of proliferating cells. The raw numbers of proliferating atrial cardiomyocytes can be found in *Figure 6—source data 1*.

### Luciferase assays

HEK293T cells were obtained from ATCC (CRL-3216) and authenticated by Short Tandem Repeat Analysis matching 100% with the database profile (ATCC Cell Line Authentication Service). Cells were tested using the MycoAlert PLUS Mycoplasma Detection Kit (Lonza, Walkersville, Inc.) and tested negative for mycoplasma contamination (ratio of 0.24). Cells were transfected with Lipofectamine 2000 (Life Technologies, Carlsbad, CA) 24 hr after seeding in a 24-well plate and were harvested 48 hr post transfection. Assays were performed in quadruplicates, and the ratio of activities of firefly luciferase (expressed from a UAS-Luc2 reporter plasmid) to Renilla luciferase was determined for each well using the Dual-Luciferase Reporter Assay System (Promega). GAL4 and Meis2-GAL4 fusions were expressed under a CMV promoter in pCS2+ vector. The following amounts of plasmid DNA were transfected per well: Renilla luciferase (pTK-RL) (50 ng), GAL4 or GAL4 fusions (500 ng), UAS-Luc2 (450 ng).

## Statistical analyses

No statistical tests were used to predetermine sample size. Several independent experiments were performed to ensure reproducibility. The investigators were inherently blinded by the experimental design during the experiment shown in *Figure 6—figure supplement 2*, until the genotyping was performed. Statistical analyses were performed by paired students t-test with one tail distribution, with the exception of the RT-qPCR analyses comparing left vs. right atria in mouse, where paired students t-test with two tailed was used. For the adult zebrafish electrocardiograms, the statistical differences were calculated using F-test and Bartlett's test. For all bar graphs, data are represented as mean ± SD. p-Values were considered significant at $p < 0.05$.

## Acknowledgements

This work was supported by the Deutsche Forschungsgemeinschaft (SFB/TR23) (S Reischauer), the Hartstichting (Dutch Heart Foundation) and the Netherlands Organization for International Cooperation in Higher Education (Nuffic) (RFV Germano), the NIH (K08HL125945) and American Heart Association (15GPSPG238300004) (Rima Arnaout), the Leducq Foundation (V. Uribe), the NIH (HL54737) and Max Planck Society (DYR Stainier).We thank Miguel Torres for providing the MEIS antibodies, Arica Beisaw for help with ChIP-qPCR experiments, Vasanth Vedantham for the use of electrocardiography equipment, Michelle Collins for discussion and reagents, Jenny Pestel, Radhan Ramadass, Simon Howard, and Hans-Martin Maischein for technical assistance.

## Additional information

### Competing interests

Didier YR Stainier: Senior Editor, eLife. The other authors declare that no competing interests exist.

### Funding

| Funder | Grant reference number | Author |
|---|---|---|
| Deutsche Forschungsge-meinschaft | SFB/TR23 | Sven Reischauer |
| National Institutes of Health | K08HL125945 | Rima Arnaout |
| American Heart Association | 15GPSPG238300004 | Rima Arnaout |
| Max-Planck-Gesellschaft | Open-access funding | Didier YR Stainier |
| National Institutes of Health | HL54737 | Didier YR Stainier |
| Hartstichting | | Raoul F. V. Germano |
| Netherlands Organization for International Cooperation in Higher Education | NL.11/379 | Raoul F. V. Germano |
| Fondation Leducq | | Verónica Uribe |

The funders had no role in study design, data collection and interpretation, or the decision to submit the work for publication.

### Author contributions

Almary Guerra, Conceptualization, Resources, Data curation, Formal analysis, Validation, Investigation, Visualization, Methodology, Writing—original draft, Writing—review and editing; Raoul FV Germano, Resources, Formal analysis, Investigation, Visualization, Methodology, Writing—review and editing; Oliver Stone, Resources, Writing—review and editing; Rima Arnaout, Benoit Vanhollebeke, Investigation, Writing—review and editing; Stefan Guenther, Data curation, Formal analysis, Investigation; Suchit Ahuja, Verónica Uribe, Resources, Established the Tg(myl7:BFP-CAAX)bns193 zebrafish transgenic line; Didier YR Stainier, Supervision, Funding acquisition, Methodology, Project administration, Writing—review and editing; Sven Reischauer, Conceptualization, Resources, Formal

analysis, Supervision, Funding acquisition, Validation, Investigation, Methodology, Project administration, Writing—review and editing

### Author ORCIDs
Almary Guerra (iD) http://orcid.org/0000-0002-5080-9559
Raoul FV Germano (iD) http://orcid.org/0000-0002-1247-0689
Didier YR Stainier (iD) http://orcid.org/0000-0002-0382-0026
Sven Reischauer (iD) http://orcid.org/0000-0002-6955-9481

### Ethics
Animal experimentation: All animal experiments were done in accordance with institutional (MPG) and national ethical and animal welfare guidelines approved by the ethics committee for animal experiments at the Regierungspräsidium Darmstadt, Germany (permit numbers B2/1041, B2-1023 and B2/Anz. 1007)

### Decision letter and Author response
Decision letter https://doi.org/10.7554/eLife.32833.043
Author response https://doi.org/10.7554/eLife.32833.044

## Additional files

### Supplementary files
• Supplementary file 1. Set of primers used to genotype *meis2b* mutant and WT alleles, and to generate *TgBAC(meis2b:Gal4FF)*[bns15]. Set of primers used to clone in situ hybridization probes. Set of primers employed for RT-qPCR and qPCR analyses.
DOI: https://doi.org/10.7554/eLife.32833.034
• Transparent reporting form
DOI: https://doi.org/10.7554/eLife.32833.035

### Major datasets
The following dataset was generated:

| Author(s) | Year | Dataset title | Dataset URL | Database, license, and accessibility information |
|---|---|---|---|---|
| Guerra A, Germano RFV, Stone O, Arnaout R, Vanhollebeke B, Guenther S, Ahuja S, Stainier DYR, Reischauer S | 2017 | Atrial Molecular Asymmetry Precedes the Emergence of Cardiac Septation | https://www.ncbi.nlm.nih.gov/geo/query/acc.cgi?acc=GSE94492 | Publicly available at the NCBI Gene Expression Omnibus (accession no: GSE94492) |

The following previously published datasets were used:

| Author(s) | Year | Dataset title | Dataset URL | Database, license, and accessibility information |
|---|---|---|---|---|
| Kahr PC, Piccini I | 2011 | Genome-wide analysis of atrial left-right differences in two different mouse strains | https://www.ncbi.nlm.nih.gov/geo/query/acc.cgi?acc=GSE29500 | Publicly available at the NCBI Gene Expression Omnibus (accession no: GSE29500) |
| Penkov D, Mateos San Martín D, Fernandez-Díaz LC, Rosselló CA | 2013 | TALE transcription factors binding and transcriptional effect | https://www.ncbi.nlm.nih.gov/geo/query/acc.cgi?acc=GSE39609 | Publicly available at the NCBI Gene Expression Omnibus (accession no: GSE39609) |

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
