## [Decision Letter]

Thank you for submitting your article "Distinct myocardial lineages break atrial symmetry during cardiogenesis in zebrafish" for consideration by *eLife*. Your article has been reviewed by two peer reviewers, and the evaluation has been overseen by a Reviewing Editor and Marianne Bronner as the Senior Editor. The reviewers have opted to remain anonymous.

The reviewers have discussed the reviews with one another and the Reviewing Editor has drafted this decision to help you prepare a revised submission.

In this interesting paper, the authors study the complex processes of cell diversification and morphogenesis that underlie cardiac left right asymmetry. They describe an unexpected laterality in the single zebrafish atrium analogous to that of the two atria in terrestrial vertebrates and demonstrate the importance of Meis2b-dependent transcription of Ptix2c in determination of this asymmetry.

The following points should be addressed before publication:

1) The authors state: "Together, these findings strongly support the existence of atrial compartmentalization in zebrafish from early stages and throughout life (summarized in Figure 4D-G') and constitute, to our knowledge, the first molecular evidence of atrial asymmetry preceding the evolutionary emergence of the inter-atrial septum of terrestrial vertebrates."

This is overstated and the latter part of the statement should be removed as Pitx2 is known to be expressed in the mouse sinus venosus prior to atrial formation. It is not necessary to make such claims.

2) The Isl1 knockdown data needs to be validated by showing reduced Isl1 expression and it should be stated if the morpholino has been validated by a genetic mutant. If not then this validation must be done.

3) The statement about current understanding of atrial laterality in mammals not requiring the SHF should be removed. This is unknown and very poorly understood.

4) The authors need to provide direct evidence that Meis2 regulates Pitx2c by ChIP Pcr experiments at minimum.

---

## [Author Response]

[…] The following points should be addressed before publication:1) The authors state: "Together, these findings strongly support the existence of atrial compartmentalization in zebrafish from early stages and throughout life (summarized in Figure 4D-G') and constitute, to our knowledge, the first molecular evidence of atrial asymmetry preceding the evolutionary emergence of the inter-atrial septum of terrestrial vertebrates."This is overstated and the latter part of the statement should be removed as Pitx2 is known to be expressed in the mouse sinus venosus prior to atrial formation. It is not necessary to make such claims.

We would like to thank the reviewers for bringing this sentence to our attention. We believe that our data including in situ hybridizations on adult hearts, confocal imaging using the *Tg(meis2b)* as well as transcriptional profiling approaches provide compelling evidence for a molecular compartmentalization of the single zebrafish atrium from early stages.

With the second part of the aforementioned sentence, we intended to illustrate our finding from an evolutionary perspective, and not a developmental perspective. Admittedly however, our initial sentence could be misleading. To clarify this point, we have rephrased the sentence (subsection “Atrial laterality appears to derive from two distinct antero-posterior progenitor fields”).

2) The Isl1 knockdown data needs to be validated by showing reduced Isl1 expression and it should be stated if the morpholino has been validated by a genetic mutant. If not then this validation must be done.

We agree that zebrafish studies relying on morpholino knock-down approaches should be treated with extreme care. In our revised manuscript, we have included a new figure with the validation of the *isl1* morpholino (please see new Figure 5—figure supplement 1). In this figure, we show that the cardiac phenotype of the *isl1* morphants recapitulates that of a previously published *isl1* mutant (1-3). Additionally, we observed that the motility of the *isl1* morphants was significantly reduced, as is that of *isl1* mutants (1). We were also able to observe the reduction of Isl1 protein in *isl1* morphants. Furthermore, to determine whether the *isl1* morphant phenotype could be due to toxic effects or non-specific targets, we injected the *isl1* morpholino in the *isl1* mutants. We did not observe any variation in phenotypes or phenotypic severity between the uninjected *isl1* mutants and the *isl1* mutants injected with the *isl1* morpholino. Altogether, these results suggest that the *isl1* morpholino effectively knocks-down Isl1 and apparently with minimal side effects.

3) The statement about current understanding of atrial laterality in mammals not requiring the SHF should be removed. This is unknown and very poorly understood.

This statement has now been removed.

4) The authors need to provide direct evidence that Meis2 regulates Pitx2c by ChIP Pcr experiments at minimum.

We agree with the reviewers that additional experimental data will help better understand the observed loss of *pitx2c* expression in the atria of *meis2b* mutants. We have now used bioinformatics tools to identify potential MEIS2 binding sites in the murine *Pitx2* locus. Furthermore, we performed ChIP-qPCR experiments on E12 embryonic mouse trunks, using antibodies that recognize MEIS1 and MEIS2. We found that MEIS1/2 bind putative regulatory elements within the *Pitx2* locus.

In detail:

In the paper of Penkov et al., 2013 (4), a ChIP-seq experiment was performed using antibodies recognizing different isoforms of MEIS1 and MEIS2. Although their work does not focus on *Pitx2*, we found in the publicly available supplementary data a putative binding site for MEIS1/2 within an annotated regulatory element located 3’ of the *Pitx2* locus, with the closest transcriptional start site belonging to *Pitx2c*. We found that this ChIP-seq peak covering 187 bp, contains a 53 bp sequence that is highly conserved amongst mammalian species (Figure 7—figure supplement 2). Importantly, this 53 bp long sequence, harbors a canonical MEIS1/2 binding motif (Figure 7D).

We have now tested these findings experimentally by performing ChIP-qPCR analysis on E12 embryonic mouse trunks of two regions of the *Pitx2* regulatory element flanking the MEIS1/2 binding motif. We also tested the *Hoxa5* locus (a known direct target of MEIS1/2 (4)) as a positive control, and *Myh6* (a region where no MEIS1/2 binding site is present) as a negative control. We observed an enrichment for *Pitx2* of 8.87 ( ± 1.92) to 10.56 ( ± 2.20) fold higher than for *Myh6*, while enrichment for the *Hoxa5* locus was 20.21 fold ( ± 7.21).

We have added these new data to the main text, as well as in Figure 7D-E, Figure 7—figure supplement 2, and Figure 7—source data 4.

References:

1) de Pater E, Clijsters L, Marques SR, Lin YF, Garavito-Aguilar ZV, Yelon D, et al. Distinct phases of cardiomyocyte differentiation regulate growth of the zebrafish heart. Development. 2009;136(10):1633-41.

2) Witzel HR, Jungblut B, Choe CP, Crump JG, Braun T, Dobreva G. The LIM protein Ajuba restricts the second heart field progenitor pool by regulating Isl1 activity. Dev Cell. 2012;23(1):58-70.

3) Hutchinson SA, Eisen JS. Islet1 and Islet2 have equivalent abilities to promote motoneuron formation and to specify motoneuron subtype identity. Development. 2006;133(11):2137-47.

4) Penkov D, Mateos San Martin D, Fernandez-Diaz LC, Rossello CA, Torroja C, Sanchez-Cabo F, et al. Analysis of the DNA-binding profile and function of TALE homeoproteins reveals their specialization and specific interactions with Hox genes/proteins. Cell Rep. 2013;3(4):1321-33.